# Fair Secretaries with Unfair Predictions

**Eric Balkanski**
Columbia University
eb3224@columbia.edu

**Will Ma**
Columbia University
wm2428@gsb.columbia.edu

**Andreas Maggiori**
Columbia University
am6292@columbia.edu

## Abstract

Algorithms with predictions is a recent framework for decision-making under uncertainty that leverages the power of machine-learned predictions without making any assumption about their quality. The goal in this framework is for algorithms to achieve an improved performance when the predictions are accurate while maintaining acceptable guarantees when the predictions are erroneous. A serious concern with algorithms that use predictions is that these predictions can be biased and, as a result, cause the algorithm to make decisions that are deemed unfair. We show that this concern manifests itself in the classical secretary problem in the learning-augmented setting—the state-of-the-art algorithm can have zero probability of accepting the best candidate, which we deem unfair, despite promising to accept a candidate whose expected value is at least $\max\{\Omega(1), 1 - O(\varepsilon)\}$ times the optimal value, where $\varepsilon$ is the prediction error. We show how to preserve this promise while also guaranteeing to accept the best candidate with probability $\Omega(1)$. Our algorithm and analysis are based on a new "pegging" idea that diverges from existing works and simplifies/unifies some of their results. Finally, we extend to the $k$-secretary problem and complement our theoretical analysis with experiments.

## 1 Introduction

As machine learning algorithms are increasingly used in socially impactful decision-making applications, the fairness of those algorithms has become a primary concern. Many algorithms deployed in recent years have been shown to be explicitly unfair or reflect bias that is present in training data. Applications where automated decision-making algorithms have been used and fairness is of central importance include loan/credit-risk evaluation [52, 40, 50], hiring [11, 17], recidivism evaluation [53, 1, 24, 15, 19], childhood welfare systems [16], job recommendations [44], price discrimination [18], resource allocation [51], and others [34, 33, 35, 8, 49]. A lot of work in recent years has been devoted to formally defining different notions of fairness [47, 38, 29, 26, 20, 42, 41], designing algorithms that satisfy these different definitions [37, 36, 12, 13, 55], and investigating trade-offs between fairness and other optimization objectives [9, 10].

While most fairness work concentrates on classification problems where the instance is known offline, we explore the problem of making fair decisions when the input is revealed in an online manner. Although fairness in online algorithms is an interesting line of research per se, fairness considerations have become increasingly important due to the recent interest in incorporating (possibly biased) machine learning predictions into the design of classical online algorithms. This framework, usually referred to as *learning-augmented algorithms* or *algorithms with predictions*, was first formalized in [48]. In contrast to classical online algorithms problems where it is assumed that no information is known about the future, learning-augmented online algorithms are given as input,

possibly erroneous, predictions about the future. The main challenge is to simultaneously achieve an improved performance when the predictions are accurate and a robust performance when the predictions are arbitrarily inaccurate. A long list of online problems have been considered in this setting and we point to [45] for an up-to-date list of papers. We enrich this active area of research by investigating how potentially biased predictions affect the fairness of decisions made by learning-augmented algorithms, and ask the following question:

> Can we design fair algorithms that take advantage of unfair predictions?

In this paper, we study this question on a parsimonious formulation of the secretary problem with predictions, motivated by fairness in hiring candidates.

**The problem.** In the classical secretary problem, there are $n$ candidates who each have a value and arrive in a random order. Upon arrival of a candidate, the algorithm observes the value of that candidate and must irrevocably decide whether to accept or reject that candidate. It can only accept one candidate and the goal is to maximize the probability of accepting the candidate with maximum value. In the classical formulation, only the *ordinal* ranks of candidates matter, and the algorithm of Dynkin [27] accepts the best candidate with a constant probability, that equals the best-possible $1/e$.

In the learning-augmented formulation of the problem proposed by Fujii and Yoshida [30], the algorithm is initially given a predicted value about each candidate and the authors focus on comparing the expected *cardinal* value accepted by the algorithm to the maximum cardinal value. The authors derive an algorithm that obtains expected value at least $\max\{\Omega(1), 1 - O(\varepsilon)\}$ times the maximum value, where $\varepsilon \geq 0$ is the prediction error. The strength of this guarantee is that it approaches 1 as the prediction error decreases and it is a positive constant even when the error is arbitrarily large.

However, because the algorithm is now using predictions that could be biased, the best candidate may no longer have any probability of being accepted. We view this as a form of unfairness, and aim to derive algorithms that are fair to the best candidate by guaranteeing them a constant probability of being accepted (we contrast with other notions of fairness in stopping problems in Section 1.1). Of course, a simple way to be fair by this metric is to ignore the predictions altogether and run the classical algorithm of Dynkin. However, this approach would ignore potentially valuable information and lose the improved guarantee of [30] that approaches 1 when the prediction error is low.

**Outline of results.** We first formally show that the algorithm of [30] may in fact accept the best candidate with 0 probability. Our main result is then a new algorithm for secretary with predictions that: obtains expected value at least $\max\{\Omega(1), 1 - O(\varepsilon)\}$ times the maximum value, like [30]; and ensures that, under any predictions, the best candidate is hired with $\Omega(1)$ probability. This result takes advantage of potentially biased predictions to achieve a guarantee on expected value that approaches 1 when the prediction error is small, while also providing a fairness guarantee for the true best candidate irrespective of the predictions. We note that Antoniadis et al. [3] also derive an algorithm for secretary with predictions, where the prediction is of the maximum value (a less informative form of prediction). This algorithm accepts the best candidate with constant probability but it does not provide a guarantee on the expected value accepted that approaches 1 as the prediction error approaches 0. Similarly, Dynkin's algorithm for the classical secretary problem accepts the best candidate with constant probability but does not make use of predictions at all.

Our algorithm is fundamentally different from existing algorithms for secretary with predictions, as our "pegging" idea, i.e., the idea not to accept a possibly suboptimal candidate if there is a future candidate with high enough predicted value, is important to achieve our fairness desideratum. We also note that the definitions of the prediction error $\varepsilon$ differ in [30] and [3]; the former error definition uses the maximum ratio over all candidates between their predicted and true value while the latter uses the absolute difference. Our techniques present an arguably simpler analysis and extend to a general family of prediction error measures that includes both of these error definitions.

We then extend our approach to the multiple choice or $k$-secretary problem where the goal is to accept at most $k$ candidates and maximize the total of their values, which is the most technical part of the paper. We design an algorithm that obtains expected total value at least $\max\{\Omega(1), 1 - O(\varepsilon)\}$ times the optimum (which is the sum of the $k$ highest values), while simultaneously guaranteeing the $k$ highest-valued candidates a constant probability of being accepted. We also have a refined guarantee that provides a higher acceptance probability for the $(1 - \delta)k$ highest-valued candidates, for any $\delta \in (0, 1)$.

Finally, we simulate our algorithms in the exact experimental setup of Fujii and Yoshida [30]. We find that they perform well both in terms of expected value accepted and fairness, whereas benchmark algorithms compromise on one of these desiderata.

## 1.1 Related work

**The secretary problem.** After Gardner [31] introduced the secretary problem, Dynkin [27] developed a simple and optimal stopping rule algorithm that, with probability at least $1/e$, accepts the candidate with maximum value. Due to its general and simple formulation, the problem has received a lot of attention (see, e.g., [46, 32] and references therein) and it was later extended to more general versions such as $k$-secretary [43], matroid-secretary [7] and knapsack-secretary [6].

**Secretaries with predictions.** The two works which are closest to our paper are those of Antoniadis et al. [3] and Fujii and Yoshida [30]. Both works design algorithms that use predictions regarding the values of the candidates to improve the performance guarantee of Dynkin's algorithm when the predictions are accurate while also maintaining robustness guarantees when the predictions are arbitrarily wrong. Antoniadis et al. [3] uses as prediction only the maximum value and defines the prediction error as the additive difference between the predicted and true maximum value while Fujii and Yoshida [30] receives a prediction for each candidate and defines the error as the maximum multiplicative difference between true and predicted value among all candidates. Very recently, Choo and Ling [14] showed that any secretary algorithm that is 1-consistent cannot achieve robustness better than $1/3 + o(1)$, even with predictions for each candidate. This result implies that, if we wish to maintain the $1 - O(\varepsilon)$ competitive ratio guarantee from [30], then the probability of accepting the best candidate cannot be better than $1/3 + o(1)$.

**Secretaries with distributional advice.** Another active line of work is to explore how *distributional* advice can be used to surpass the $1/e$ barrier of the classical secretary problem. Examples of this line of work include the *prophet secretary problems* where each candidate draws its valuation from a known distribution [28, 22, 23, 5] and the *sample secretary problem* where the algorithm designer has only sample access to this distribution [39, 21]. We note that in the former models, predictions are either samples from distributions or distributions themselves which are assumed to be perfectly correct, while in the learning-augmented setting, we receive point predictions that could be completely incorrect. Dütting et al. [25] investigate a general model for advice where both values and advice are revealed upon a candidate's arrival and are drawn from a joint distribution $\mathcal{F}$. For example, their advice can be a noisy binary prediction about whether the current candidate is the best overall. Their main result uses linear programming to design optimal algorithms for a broad family of advice that satisfies two conditions. However, these two conditions are not satisfied by the predictions we consider. Additionally, we do not assume any prior knowledge of the prediction quality, whereas their noisy binary prediction setting assumes that the error probability of the binary advice is known.

**Fairness in stopping algorithms.** We say that a learning-augmented algorithm for the secretary problem is $F$-fair if it accepts the candidate with the maximum true value with probability at least $F$. In that definition, we do not quantify unfairness as a prediction property but as an algorithmic one, since the algorithm has to accept the best candidate with probability at least $F$ no matter how biased predictions are our fairness notion is a challenging one. That notion can be characterized as an individual fairness notion similar to the *identity-independent fairness* (IIF) and *time-independent fairness* (TIF) introduced in [4]. In the context of the secretary problem, IIF and TIF try to mitigate discrimination due to a person's identity and arrival time respectively. While these are very appealing fairness notions, the fair algorithms designed in [4] fall in the classical online algorithms setting as they do not make any assumptions about the future. Consequently, their performance is upper bound by the performance of the best algorithm in the classical worst-case analysis setting. It is also interesting to note the similarities with the *poset secretary problem* in [54]. In the latter work the set of candidates is split into several groups and candidates belonging to different groups cannot be compared due to different biases in the evaluation. In some sense, we try to do the same; different groups of candidates may have predictions that are affected by different biases making the comparison difficult before the true value of each candidate is revealed. Again, in [54] no information about the values of future candidates is available and the performance of their algorithms is upper bounded by the best possible performance in the worst-case analysis setting.

## 2 Preliminaries

**Secretary problem with predictions.** Candidates $i = 1, \ldots, n$ have true values $u_i$ and predicted values $\hat{u}_i$. The number of candidates $n$ and their predicted values are known in advance. The candidates arrive in a uniformly random order. Every time a new candidate arrives their true value is revealed and the algorithm must immediately decide whether to accept the current candidate or reject them irrevocably and wait for the next arrival. We let $i^* = \operatorname{argmax}_i u_i$ and $\hat{i} = \operatorname{argmax}_i \hat{u}_i$ denote the indices of the candidates with the maximum true and predicted value respectively. An instance $\mathcal{I}$ consists of the $2n$ values $u_1, \ldots, u_n, \hat{u}_1, \ldots, \hat{u}_n$ which, for convenience, are assumed to be positive[1] and mutually distinct[2]. We let $\varepsilon(\mathcal{I})$ denote its *prediction error*. For simplicity, we focus on the additive prediction error $\varepsilon(\mathcal{I}) = \max_i |\hat{u}_i - u_i|$, but we consider an abstract generalization that includes the multiplicative prediction error of [30] in Appendix A.2.

**Objectives.** We let $\mathcal{A}$ be a random variable denoting the candidate accepted by a given algorithm on a fixed instance, which depends on both the arrival order and any internal randomness in the algorithm. We consider the following desiderata for a given algorithm:

$$\mathbf{E}[u_{\mathcal{A}}] \geq u_{i^*} - C \cdot \varepsilon(\mathcal{I}), \ \forall \mathcal{I} \qquad \textit{(smoothness)}$$
$$P[\mathcal{A} = i^*] \geq F, \ \forall \mathcal{I}. \qquad \textit{(fairness)}$$

Since the prediction error $\varepsilon(\mathcal{I})$ is an additive prediction error, we define smoothness to provide an additive approximation guarantee that depends on $\varepsilon(\mathcal{I})$. When considering the multiplicative prediction error of [30], smoothness is defined to provide an approximation guarantee that is multiplicative instead of additive (see Theorem 4).

We aim to derive algorithms that can satisfy smoothness and fairness with constants $C, F > 0$ that do not depend on the instance $\mathcal{I}$ or the number of candidates $n$. Existing algorithms for secretary with predictions do not simultaneously satisfy these desiderata, as shown by our examples in Appendix A.1.

**Comparison to other objectives.** Existing algorithms for secretary with predictions do satisfy a weaker notion called *R-robustness*, where $\mathbf{E}[u_{\mathcal{A}}] \geq R \cdot u_{i^*}$ for some constant $R > 0$. Our desideratum of fairness implies $F$-robustness and aligns with the classical secretary formulation where one is only rewarded for accepting the best candidate. Another notion of interest in existing literature is *consistency*, which is how $\mathbf{E}[u_{\mathcal{A}}]$ compares to $u_{i^*}$ when $\varepsilon(\mathcal{I}) = 0$. Our smoothness desideratum implies 1-consistency, the best possible consistency result, and guarantees a smooth degradation as $\varepsilon(\mathcal{I})$ increases beyond 0.

## 3 Algorithm and Analysis

We first present and analyze ADDITIVE-PEGGING in Algorithm 1 which achieves the desiderata from Section 2. Then, we mention how using a more abstract prediction error and an almost identical analysis, permits us to generalize ADDITIVE-PEGGING to PEGGING which achieves comparable guarantees for a more general class of error functions that includes the multiplicative error.

Our algorithms assume that each candidate $i$ arrives at an independently random arrival time $t_i$ drawn uniformly from $[0, 1]$. The latter continuous-time arrival model is equivalent to candidates arriving in a uniformly random order and simplifies the algorithm description and analysis. We also write $\epsilon_i$ as shorthand for $|u_i - \hat{u}_i|$, $\varepsilon$ as shorthand for $\varepsilon(\mathcal{I})$ (so that $\varepsilon = \max_i \epsilon_i$) and $i \prec j$ if $t_i < t_j$.

**Description of ADDITIVE-PEGGING.** ADDITIVE-PEGGING ensures smoothness by always accepting a candidate whose value is close to $u_{\hat{i}}$ which, as we argue, is at least $u_{i^*} - 2\varepsilon$. To see this, note that $u_{\hat{i}} \geq \hat{u}_{\hat{i}} - \epsilon_{\hat{i}} \geq \hat{u}_{i^*} - \epsilon_{\hat{i}} \geq u_{i^*} - \epsilon_{i^*} - \epsilon_{\hat{i}} \geq u_{i^*} - 2\varepsilon$, where we used that $\hat{u}_{\hat{i}} \geq \hat{u}_{i^*}$ (by definition of $\hat{i}$) and $\varepsilon \geq \max\{\epsilon_{i^*}, \epsilon_{\hat{i}}\}$ (by definition of $\varepsilon$). Consequently, for smoothness, our algorithm defines the literal $\mathcal{C} = (i = \hat{i})$ at each new arrival, which is true if and only if $i$ has the highest predicted value. Accepting while $\mathcal{C}$ holds would maintain smoothness.

For the fairness desideratum, we note that Dynkin's algorithm [27] for the classical secretary problem relies on the observation that if a constant fraction of the candidates have arrived and the candidate

---

[1]Our results for additive error allow negative values, but our extension to multiplicative error in Appendix A.2 requires positive values.

[2]This is without loss as adding an arbitrarily small perturbation to each true and predicted value does not change the performance of our algorithms. This allows for a unique $\operatorname{argmax}$ in the definitions of $i^*$ and $\hat{i}$.

who just arrived has the maximum true value so far, then this candidate has a constant probability of being the best overall. The same high-level intuition is used in our algorithm. Every time a new candidate $i$ arrives, we check if $i$ is the maximum so far and if $t_i > 1/2$; namely, we compute the literal $\mathcal{F}$. Accepting when $\mathcal{F}$ is true, which is what Dynkin's algorithm does, would ensure fairness.

However, there are two crucial situations where ADDITIVE-PEGGING differs from Dynkin's algorithm. The first such situation is when the candidate $\hat{i}$ with maximum predicted value arrives and we have that $\hat{i}$ is not the maximum so far or $t_{\hat{i}} \leq 1/2$, i.e., $\mathcal{C} \wedge \overline{\mathcal{F}}$ is true. In this case, we cannot always reject $\hat{i}$, as Dynkin's algorithm would, because that would not guarantee smoothness. Instead, we reject $\hat{i}$ only if there is a future candidate whose prediction is sufficiently high compared to $u_{\hat{i}}$. We call $I^{\text{pegged}}$ the set of those candidates. The main idea behind the pegged set $I^{\text{pegged}}$ is that it contains the last candidate to arrive who can guarantee the smoothness property, which is why we accept that candidate when they arrive. The second situation where our algorithm departs from Dynkin's algorithm is when a candidate $i$ arrives with $i \neq \hat{i}, i \neq i^{\text{pegged}}$ and we have that $\mathcal{F}$ is true, in which case Algorithm 1 executes the if statement under the case $\overline{\mathcal{C}} \wedge \mathcal{F}$. In this situation, we cannot always accept $i$ as Dynkin's algorithm would, because that would again violate smoothness. Instead, we accept $i$ only if $u_i$ can be lower bounded by $\hat{u}_{\hat{i}} - \varepsilon_{t_i}$, noting that if conversely $u_i + \varepsilon_{t_i}$ is smaller than $\hat{u}_{\hat{i}}$, then accepting $i$ might be detrimental to our quest of ensuring smoothness.

---

**Algorithm 1** ADDITIVE-PEGGING

//\* The algorithm stops when it accepts a candidate by executing $\mathcal{A} \leftarrow i$. \*//
**Initialization:** $I^{\text{pegged}} \leftarrow \emptyset$
**while** agent $i$ arrives at time $t_i$ **do**
    **if** $i \in I^{\text{pegged}}$ **then**
        **if** $|I^{\text{pegged}}| = 1$ **then**
            $\mathcal{A} \leftarrow i$
        **else**
            $I^{\text{pegged}} \leftarrow I^{\text{pegged}} \setminus \{i\}$
    $\mathcal{F} \leftarrow (u_i > \max_{j \prec i} u_j) \wedge (t_i > 1/2), \mathcal{C} \leftarrow (i = \hat{i}), \varepsilon_{t_i} \leftarrow \max_{j:t_j \leq t_i} |\hat{u}_j - u_j|$
    **if** $\mathcal{C} \wedge \mathcal{F}$ **then**
        $\mathcal{A} \leftarrow i$
    **else if** $\mathcal{C} \wedge \overline{\mathcal{F}}$ **then**
        $I^{\text{pegged}} \leftarrow \{j \succ \hat{i} : u_{\hat{i}} < \hat{u}_j + \varepsilon_{t_i}\}$ (note that $\hat{i} = i$)
        **if** $I^{\text{pegged}} = \emptyset$ **then**
            $\mathcal{A} \leftarrow i$
    **else if** $\overline{\mathcal{C}} \wedge \mathcal{F}$ **then**
        **if** $u_i > \hat{u}_{\hat{i}} - \varepsilon_{t_i}$ **then**
            $\mathcal{A} \leftarrow i$

---

**Analysis of the ADDITIVE-PEGGING algorithm.**

**Lemma 1.** ADDITIVE-PEGGING *satisfies* $u_{\mathcal{A}} \geq u_{i^*} - 4\,\varepsilon(\mathcal{I})$, $\forall \mathcal{I}$ *with probability 1.*

*Proof.* Let $i^{\text{pegged}}$ denote the last arriving candidate in $I^{\text{pegged}}$.

We first argue that PEGGING always accepts a candidate irrespective of the random arrival times of the candidates. We focus on any instance where ADDITIVE-PEGGING does not accept a candidate until time $t_{\hat{i}}$. At time $t_{\hat{i}}$ either $\mathcal{C} \wedge \mathcal{F}$ or $\mathcal{C} \wedge \overline{\mathcal{F}}$ are true. Since in the former case candidate $\hat{i}$ is accepted, we focus on the latter case and in particular whenever the set $I^{\text{pegged}}$ which is computed is non-empty (otherwise, candidate $\hat{i}$ is accepted). In that case, it is guaranteed that by time $t_{i^{\text{pegged}}}$ ADDITIVE-PEGGING will accept a candidate.

We now argue that in all cases ADDITIVE-PEGGING maintains smoothness. Using $\varepsilon$, $\epsilon_i$ definitions and the fact that $\hat{i}$ is the candidate with the maximum predicted value we have: $\hat{u}_{\hat{i}} \geq \hat{u}_{i^*} \geq u_{i^*} - \epsilon_{i^*} \geq u_{i^*} - \varepsilon$. If candidate $\hat{i}$ is accepted then using the latter lower bound we get $u_{\hat{i}} \geq \hat{u}_{\hat{i}} - \epsilon_{\hat{i}} \geq u_{i^*} - \varepsilon - \epsilon_{\hat{i}} \geq u_{i^*} - 2\,\varepsilon$. If we accept $i \neq \hat{i}$ and the if statement of $\overline{\mathcal{C}} \wedge \mathcal{F}$ is executed at time $t_i$ then we have $u_i > \hat{u}_{\hat{i}} - \varepsilon_{t_i} \geq u_{i^*} - \varepsilon - \varepsilon_{t_i} \geq u_{i^*} - 2\,\varepsilon$. Finally, we need to lower bound the value $u_{i^{\text{pegged}}}$ in case our algorithm terminates accepting $i^{\text{pegged}}$. Note that from the way

the pegged set $I^{\text{pegged}}$ is updated when $\mathcal{C} \wedge \overline{\mathcal{F}}$ is true we always have $u_{\hat{\imath}} < \hat{u}_{i^{\text{pegged}}} + \varepsilon_{t_{\hat{\imath}}}$. Since $u_{i^{\text{pegged}}} \geq \hat{u}_{i^{\text{pegged}}} - \epsilon_{i^{\text{pegged}}}$ we can conclude that $u_{i^{\text{pegged}}} > u_{\hat{\imath}} - \varepsilon_{t_{\hat{\imath}}} - \epsilon_{i^{\text{pegged}}} \geq u_{i^*} - 4\varepsilon$. $\qquad\square$

**Lemma 2.** ADDITIVE-PEGGING *satisfies* $P[\mathcal{A} = i^*] \geq 1/16$, $\forall \mathcal{I}$.

*Proof.* In the following, we assume that the number of candidates is larger or equal to 3. The proof for the case where $n = 2$ is almost identical while the fairness guarantee in that case is $1/4$. We denote by $\tilde{\imath}$ the index of the candidate with the highest true value except $i^*$ and $\hat{\imath}$, i.e., $\tilde{\imath} = \text{argmax}_{i \neq i^*, \hat{\imath}} u_i$. Note that depending on the value of $u_{\hat{\imath}}$, $\tilde{\imath}$ might denote the index of the candidate with the second or third highest true value. To prove fairness we distinguish between two cases: either $\hat{\imath} = i^*$ or $\hat{\imath} \neq i^*$. For each of those cases, we define an event and argue that: (1) the event happens with constant probability, and (2) if that event happens then ADDITIVE-PEGGING accepts $i^*$.

If $i^* = \hat{\imath}$ we define event $E = \{t_{\tilde{\imath}} < 1/2 < t_{i^*}\}$ for which $P[E] = 1/4$. $E$ implies that our algorithm does not accept any candidate until time $t_{i^*}$. Indeed, note that at any point in time before $t_{i^*}$, both literals $\mathcal{F}$ and $\mathcal{C}$ are simultaneously false. On the contrary, at time $t_{i^*}$, both $\mathcal{C}$ and $\mathcal{F}$ are true and our algorithm accepts $i^*$.

On the other hand, if $i^* \neq \hat{\imath}$ we distinguish between two sub-cases. First, we show that either $u_{\hat{\imath}} < \hat{u}_{i^*} + \epsilon_{\hat{\imath}}$ or $u_{i^*} > \hat{u}_{\hat{\imath}} - \epsilon_{i^*}$ is true. By contradiction, assume that both inequalities do not hold, then

$$u_{\hat{\imath}} \geq \hat{u}_{i^*} + \epsilon_{\hat{\imath}} \xrightarrow{u_{i^*} > u_{\hat{\imath}}} u_{i^*} > \hat{u}_{i^*} + \epsilon_{\hat{\imath}} \Rightarrow u_{i^*} - \hat{u}_{i^*} > \epsilon_{\hat{\imath}} \xrightarrow{\epsilon_{i^*} \geq u_{i^*} - \hat{u}_{i^*}} \epsilon_{i^*} > \epsilon_{\hat{\imath}}$$

$$u_{i^*} \leq \hat{u}_{\hat{\imath}} - \epsilon_{i^*} \xrightarrow{u_{i^*} > u_{\hat{\imath}}} u_{\hat{\imath}} < \hat{u}_{\hat{\imath}} - \epsilon_{i^*} \Rightarrow \epsilon_{i^*} < \hat{u}_{\hat{\imath}} - u_{\hat{\imath}} \xrightarrow{\epsilon_{\hat{\imath}} \geq u_{\hat{\imath}} - \hat{u}_{\hat{\imath}}} \epsilon_{i^*} < \epsilon_{\hat{\imath}}$$

which is a contradiction. We now define two events $E_1$ and $E_2$ which imply that $i^*$ is always accepted whenever $\{u_{\hat{\imath}} < \hat{u}_{i^*} + \epsilon_{\hat{\imath}}\}$ and $\{u_{i^*} > \hat{u}_{\hat{\imath}} - \epsilon_{i^*}\}$ are true respectively.

If $u_{\hat{\imath}} < \hat{u}_{i^*} + \epsilon_{\hat{\imath}}$, then we define event $E_1 = \{t_{\tilde{\imath}} < 1/2\} \wedge \{t_{\hat{\imath}} < 1/2\} \wedge \{1/2 < t_{i^*}\}$ which is composed by 3 independent events and it happens with probability $P[E_1] = 1/2^3 = 1/8$. $E_1$ implies that $t_{\hat{\imath}} < t_{i^*} \Rightarrow \varepsilon_{t_{i^*}} \geq \epsilon_{\hat{\imath}}$, thus we can deduce that $u_{i^*} > u_{\hat{\imath}} \geq \hat{u}_{\hat{\imath}} - \epsilon_{\hat{\imath}} \geq \hat{u}_{\hat{\imath}} - \varepsilon_{t_{i^*}}$. Consequently, if until time $t_{i^*}$ all candidates are rejected, $E_1$ implies that $\overline{\mathcal{C}} \wedge \mathcal{F} \wedge \{u_{i^*} > \hat{u}_{\hat{\imath}} - \epsilon_{i^*}\}$ is true at time $t_{i^*}$ and candidate $i^*$ is hired. To argue that no candidate is accepted before time $t_{i^*}$, note that $\mathcal{F}$ is false at all times before $t_{i^*}$ and at time $t_{\hat{\imath}}$ (when literal $\mathcal{C}$ is true) the set $\{j \succ \hat{\imath} : u_{\hat{\imath}} < \hat{u}_j + \varepsilon_{t_{\hat{\imath}}}\} \supseteq \{j \succ \hat{\imath} : u_{\hat{\imath}} < \hat{u}_j + \epsilon_{\hat{\imath}}\}$ contains $i^*$.

If $u_{i^*} > \hat{u}_{\hat{\imath}} - \epsilon_{i^*}$, then we define $E_2 = \{t_{\tilde{\imath}} < 1/2 < t_{i^*} < t_{\hat{\imath}}\}$ which happens with probability

$$
\begin{aligned}
P[E_2] &= P[t_{\tilde{\imath}} < 1/2] \cdot P[1/2 < t_{i^*} < t_{\hat{\imath}}] \\
&= P[t_{\tilde{\imath}} < 1/2] \cdot P[1/2 < \min\{t_{i^*}, t_{\hat{\imath}}\} \wedge \min\{t_{i^*}, t_{\hat{\imath}}\} = t_{i^*}] \\
&= P[t_{\tilde{\imath}} < 1/2] \cdot P[1/2 < \min\{t_{i^*}, t_{\hat{\imath}}\}] \cdot P[\min\{t_{i^*}, t_{\hat{\imath}}\} = t_{i^*}] \\
&= (1/2) \cdot (1/4) \cdot (1/2) = 1/16
\end{aligned}
$$

Note that until time $t_{i^*}$ no candidate is accepted since $\mathcal{C}$ and $\mathcal{F}$ are both false at all times. Indeed, between times 0 and $1/2$ only $\hat{\imath}$ could have been accepted but its arrival time is after $t_{i^*}$, and between times $1/2$ and $t_{i^*}$ no candidate has a true value larger than $u_{\tilde{\imath}}$. Finally, note that at time $t_{i^*}$ we have $\varepsilon_{t_{i^*}} \geq \epsilon_{i^*}$ and consequently $\overline{\mathcal{C}} \wedge \mathcal{F} \wedge \{u_{i^*} > \hat{u}_{\hat{\imath}} - \varepsilon_{t_{i^*}}\}$ is true and $i^*$ gets accepted. $\qquad\square$

**Theorem 3.** ADDITIVE-PEGGING *satisfies smoothness and fairness with* $C = 4$ *and* $F = 1/16$.

Theorem 3 follows directly from Lemmas 1 and 2. We note that Lemma 1 actually implies a stronger notion of smoothness that holds with probability 1.

**The general PEGGING algorithm.** In Appendix A.2 we generalize the ADDITIVE-PEGGING algorithm to the PEGGING algorithm to provide fair and smooth algorithms for different prediction error definitions. ADDITIVE-PEGGING is an instantiation of PEGGING when the prediction error is defined as the maximum absolute difference between true and predicted values among candidates. To further demonstrate the generality of PEGGING, we also instantiate it over the same prediction error definition $\varepsilon(\mathcal{I}) = \max_i |1 - \hat{u}_i/u_i|$ as in [30] and recover similar smoothness bounds while also ensuring fairness. We name the latter instantiation MULTIPLICATIVE-PEGGING and present its guarantees in Theorem 4.

**Theorem 4.** *Let* $\varepsilon(\mathcal{I}) = \max_i |1 - \hat{u}_i/u_i|$ *and assume* $u_i, \hat{u}_i > 0 \ \forall i \in [n]$. *Then* MULTIPLICATIVE-PEGGING *satisfies fairness with* $F = 1/16$ *and selects a candidate* $\mathcal{A}$ *such that* $u_{\mathcal{A}} \geq u_{i^*} \cdot (1 - 4 \cdot \varepsilon(\mathcal{I}))$ *with probability 1.*

Fujii and Yoshida [30] define the prediction error as in Theorem 4 and design an algorithm that accepts a candidate with expected value at least $u_{i^*} \cdot \max\{(1 - \varepsilon)/(1 + \varepsilon), 0.215\}$. Since $(1 - \varepsilon)/(1 + \varepsilon) \geq 1 - 2\varepsilon$ their algorithm satisfies a smoothness desideratum similar to the one in Theorem 4, but as we prove in Appendix A.1, it violates the fairness desideratum.

## 4 Extension: $k$-Secretary problem with predictions

We consider the generalization to the $k$-secretary problem, where $k \geq 1$ candidates can be accepted. To simplify notation we label the candidates in decreasing order of predicted value, so that $\hat{u}_1 > \cdots > \hat{u}_n$ and denote $r_\ell$ to be the index of the candidate with the $\ell$'th highest true value so that $u_{r_1} > \cdots > u_{r_n}$. The prediction error is again defined as $\varepsilon(\mathcal{I}) := \max_i |u_i - \hat{u}_i|$ and we let $S$ denote the random set of candidates accepted by a given algorithm on a fixed instance. The extension of our two objectives to this setting is

$$\mathbf{E}\left[\sum_{i \in S} u_i\right] \geq \sum_{\ell=1}^{k} u_{r_\ell} - C \cdot \varepsilon(\mathcal{I}), \ \forall \mathcal{I} \qquad \textit{(smoothness for k-secretary)}$$

$$P[r_\ell \in S] \geq F_\ell, \ \forall \ell = 1, \dots, k, \ \ \forall \mathcal{I}. \qquad \textit{(fairness for k-secretary)}$$

The smoothness desideratum compares the expected sum of true values accepted by the algorithm to the sum of the $k$ highest true values that could have been accepted. The fairness desideratum guarantees each of the candidates ranked $\ell = 1, \dots, k$ to be accepted with probability $F_\ell$. The $k$-secretary problem with predictions has been studied by Fujii and Yoshida [30], who derive an algorithm satisfying $\mathbf{E}\left[\sum_{i \in S} u_i\right] \geq \max\{1 - O(\log k/\sqrt{k}), 1 - O(\max_i |1 - \hat{u}_i/u_i|)\} \sum_{\ell=1}^{k} u_{r_\ell}$ but without any fairness guarantees. We derive an algorithm $k$-PEGGING that satisfies the following.

**Theorem 5.** $k$-PEGGING *satisfies smoothness and fairness for $k$-secretary with* $C = 4k$ *and* $F_\ell = \max\left\{(1/3)^{k+5}, \frac{1-(\ell+13)/k}{256}\right\}$ *for all* $\ell = 1, \dots, k$.

We note that the algorithm of Kleinberg [43] for $k$-Secretary (without predictions) obtains in expectation at least

$$\left(1 - \frac{5}{\sqrt{k}}\right) \sum_{\ell=1}^{k} u_{r_\ell} \geq \sum_{\ell=1}^{k} u_{r_\ell} - 5\sqrt{k} \max_i u_i,$$

which has a better asymptotic dependence on $k$ than our smoothness constant $C = 4k$ if the prediction error is relatively large, i.e., $\varepsilon(\mathcal{I}) = \omega\left(\max_i u_i/\sqrt{k}\right)$. On the other hand, regarding our fairness guarantee, if one only considers our fairness desideratum for $k$-secretary, then a simple algorithm suffices to achieve $F_\ell = 1/4$ for all $\ell = 1, \dots, k$, namely: reject all candidates $i$ with $t_i < 1/2$; accept any candidate $i$ with $t_i > 1/2$ whose true value $u_i$ is among the $k$ highest true values observed so far, space permitting. For any of the top $k$ candidates, i.e., candidate $r_\ell$ with $\ell \in [k]$, their value $u_{r_\ell}$ is always greater than the threshold $\tau$, which our algorithm recomputes upon the arrival of each candidate. Consequently, candidate $r_\ell$ is added to our solution if the following conditions are satisfied: (1) $t_{r_\ell} > 1/2$; and (2) there is space available in the solution when $r_\ell$ arrives. For condition (2) to hold, it suffices that at least $k$ of the $2k - 1$ candidates with the highest values other than $r_\ell$ arrive before time $1/2$. This ensures that at most $k - 1$ candidates other than $r_\ell$ can be accepted after time $1/2$. The probability of both conditions (1) and (2) being satisfied is at least $P\left[t_{r_\ell} > 1/2 \wedge \text{Binom}(2k - 1, \frac{1}{2})\right] = P[t_{r_\ell} > 1/2] \cdot P\left[\text{Binom}(2k - 1, \frac{1}{2}) \geq k\right] = \frac{1}{2} \cdot \frac{1}{2} = \frac{1}{4}$, establishing that $F_\ell = \frac{1}{4}$ for all $\ell = 1, \dots, k$.

Assuming $k$ is a constant, $C$ and $F_1, \dots, F_k$ in Theorem 5 are constants that do not depend on $n$ or the instance $\mathcal{I}$. For large values of $k$ the first term in $F_\ell$ is exponentially decaying, but the second term still guarantees candidate $r_\ell$ a probability of acceptance that is independent of $k$ as long as $(\ell + 13)/k$ is bounded away from 1. More precisely, for $k \geq 52$ and $l \leq k/2$ we have that candidate $r_\ell$ is accepted with probability at least $\frac{1}{1024}$, i.e., every candidate among the top $k/2$ is accepted

with probability at least $\frac{1}{1024}$, thus $\mathbf{E}\left[\sum_{i\in S} u_i\right] \geq \frac{1}{1024}\sum_{\ell=1}^{k/2} u_{r_\ell} \geq \frac{1}{2048}\sum_{\ell=1}^{k} u_{r_\ell}$. This implies a multiplicative guarantee on total value that does not depend on $k$ when $k$ is large.

**The algorithm.** While we defer the proof of Theorem 5 to Appendix B, we present the intuition and the main technical difficulties in the design of $k$-PEGGING. The algorithm maintains in an online manner the following sets: (1) the solution set $S$ which contains all the candidates that have already been accepted; (2) a set $H$ that we call the "Hopefuls" and contains the $k - |S|$ future candidates with highest predicted values; (3) a set $B$ that we call the blaming set, which contains a subset of already arrived candidates that pegged a future candidate; and (4) the set $P$ of pegged elements which contains all candidates that have been pegged by a candidate in $B$. In addition, we use function $\mathrm{peg}$ to store the "pegging responsibility", i.e., if $\mathrm{peg}(i) = j$, for some candidates $i, j$ where $i$ had one of the $k$ highest predicted values, then $i$ was not accepted at the time of its arrival and pegged $j$. We use $\mathrm{peg}^{-1}(j) = i$ to denote that $j$ was pegged by $i$.

---

**Algorithm 2** $k$-PEGGING

---

//* The algorithm stops when it accepts $k$ candidates, i.e., when $|S| = k$. *//
**Initialization:** $H \leftarrow [k], S \leftarrow \emptyset, P \leftarrow \emptyset, B \leftarrow \emptyset$
**while** agent $i$ arrives at time $t_i$ **do**
    **if** $i \in P$ **then**                                                                  $\triangleright$ Case 1
        Add $i$ to $S$, remove $i$ from $P$, and remove $\mathrm{peg}^{-1}(i)$ from $B$
    $\tau \leftarrow k^{th}$ highest true value strictly before $t_i$
    $\mathcal{F} \leftarrow (u_i > \tau) \wedge (t_i > 1/2), \mathcal{C} \leftarrow (i \in H), \varepsilon_{t_i} \leftarrow \max_{j:t_j \leq t_i}|\hat{u}_j - u_j|$
    **if** $\mathcal{C} \wedge \mathcal{F}$ **then**                                                   $\triangleright$ Case 2
        Add $i$ to $S$ and remove $i$ from $H$
    **else if** $\mathcal{C} \wedge \overline{\mathcal{F}}$ **then**                                         $\triangleright$ Case 3
        **if** $\{j \succ i : u_i < \hat{u}_j + \varepsilon_{t_i}\} \setminus (P \cup [k]) = \emptyset$ **then**         $\triangleright$ subcase a
            Add $i$ to $S$ and remove $i$ from $H$
        **else**                                                 $\triangleright$ subcase b
            $j' \leftarrow$ An arbitrary candidate from $\{j \succ i : u_i < \hat{u}_j + \varepsilon_{t_i}\} \setminus (P \cup [k])$
            Add $j'$ to $P$, add $i$ to $B$, remove $i$ from $H$, and set $\mathrm{peg}(i) = j'$
    **else if** $\overline{\mathcal{C}} \wedge \mathcal{F}$ **then**                                           $\triangleright$ Case 4
        **if** $\{j \in B : u_i > u_j\} \neq \emptyset$ **then**                             $\triangleright$ subcase a
            $j' \leftarrow$ An arbitrary candidate from $\{j \in B : u_i > u_j\}$
            Add $i$ to $S$, remove $j'$ from $B$, and remove $\mathrm{peg}(j')$ from $P$
        **else if** $\{j \in H : u_i > \hat{u}_j - \varepsilon_{t_i}\} \neq \emptyset$ **then**          $\triangleright$ subcase b
            $j' \leftarrow$ An arbitrary candidate from $\{j \in H : u_i > \hat{u}_j - \varepsilon_{t_i}\}$
            Add $i$ to $S$ and remove $j'$ from $H$

---

To satisfy the fairness property, we check if the current candidate $i$ has arrived at time $t_i > 1/2$ and if $u_i$ is larger than the $k^{th}$ highest value seen so far. We refer to these two conditions as the fairness conditions. If $i \in P$ (*case 1*) or $i \in H$ and the fairness conditions hold (*case 2*), then we accept $i$. If the fairness conditions hold but $i \notin H$ then we accept if there is a past candidate in $B$ with lower true value than $u_i$ (*subcase 4a*), or a future candidate in $H$ with low predicted value compared to $u_i$ (*subcase 4b*). The main technical challenge in generalizing the pegging idea to $k > 1$ arises when a candidate $i \in H$ arrives, but the fairness conditions do not hold (*case 3*). In this situation, it is unclear whether to reject $i$ and peg a future candidate, or accept $i$. For instance, consider a scenario where the prediction error is consistently large (i.e., $\varepsilon_{t_i}$ is always large), such that when $i$ arrives, the set $\{j \succ i : u_i < \hat{u}_j + \varepsilon_{t_i}\} \setminus (P \cup [k])$ is always non-empty. If $t_i < 1/2$ and we accept $i$, we risk depleting our budget too quickly before time $1/2$, leaving insufficient capacity to accept candidates not in $[k]$ who arrive later. Conversely, if we reject $i$, we deny it the possibility of acceptance in the first half of the time horizon, potentially reducing its overall acceptance probability. $k$-PEGGING balances this tradeoff while achieving smoothness. To establish smoothness, we demonstrate that the $k$ candidates with the highest predicted values can be mapped to the solution set $S$, ensuring that the true values within our solution set are pairwise "close" to the values of candidates in $\{1, 2, \ldots, k\}$. This is proven in Lemma 8 by constructing an injective function $\mathrm{m}(\cdot)$ from set $S$ to $\{1, 2, \ldots, k\}$ such that for each $j \in S$, $u_j \approx u_{\mathrm{m}(j)}$.

# 5 Experiments

We simulate our ADDITIVE-PEGGING and MULTIPLICATIVE-PEGGING algorithms in the exact experimental setup of Fujii and Yoshida [30], to test its average-case performance.

**Experimental Setup.** Fujii and Yoshida [30] generate various types of instances. We follow their Almost-constant, Uniform, and Adversarial types of instances, and also create the Unfair type of instance to further highlight how slightly biased predictions can lead to very unfair outcomes. Both true and predicted values of candidates in all these instance types are parameterized by a scalar $\varepsilon \in [0, 1)$ which controls the prediction error. Setting $\varepsilon = 0$ creates instances with perfect predictions and setting a higher value of $\varepsilon$ creates instances with more erroneous predictions. Almost-constant models a situation where one candidate has a true value of $1/(1 - \varepsilon)$ and the rest of the candidates have a value of $1$. All predictions are set to $1$. In Uniform, we sample each $u_i$ independently from the exponential distribution with parameter $1$. The exponential distribution generates a large value with a small probability and consequently models a situation where one candidate is significantly better than the rest. All predicted values are generated by perturbing the actual value with the uniform distribution, i.e., $\hat{u}_i = \delta_i \cdot u_i$, where $\delta_i$ is sampled uniformly and independently from $[1 - \varepsilon, 1 + \varepsilon]$. In Adversarial, the true values are again independent samples from the exponential distribution with parameter $1$. The predictions are "adversarially" perturbed while maintaining the error to be at most $\varepsilon$ in the following manner: if $i$ belongs to the top half of candidates in terms of true value, then $\hat{u}_i = (1 - \varepsilon) \cdot u_i$; if $i$ belongs to the bottom half, then $\hat{u}_i = (1 + \varepsilon) \cdot u_i$. Finally, in Unfair all candidates have values that are at most a $(1 + \varepsilon)$ multiplicative factor apart. Formally, $u_i$ is a uniform value in $[1 - \varepsilon/4, 1 + \varepsilon/4]$, and since $(1 + \varepsilon/4)/(1 - \varepsilon/4) \leq (1 + \varepsilon)$ we have that the smallest and largest value are indeed very close. We set $\hat{u}_i = u_{n-r(i)+1}$ where $r(i)$ is the rank of $u_i$, i.e., predictions create a completely inverted order.

We compare ADDITIVE-PEGGING and MULTIPLICATIVE-PEGGING against LEARNED-DYNKIN [30], HIGHEST-PREDICTION which always accepts the candidate with the highest prediction, and the classical DYNKIN algorithm which does not use the predictions. Following [30], we set the number of candidates to be $n = 100$. We experiment with all values of $\varepsilon$ in $\{0, 1/20, 2/20, \ldots, 19/20\}$. For each type of instance and value of $\varepsilon$ in this set, we randomly generate 10000 instances, and then run each algorithm on each instance. For each algorithm, we consider instance-wise the ratio of the true value it accepted to the maximum true value, calling the average of this ratio across the 10000 instances its *competitive ratio*. For each algorithm, we consider the fraction of the 10000 instances on which it successfully accepted the candidate with the highest true value, calling this fraction its *fairness*. We report the competitive ratio and fairness of each algorithm, for each type of instance and each value of $\varepsilon$, in Figure 1. Our code is written in Python 3.11.5 and we conduct experiments on an M3 Pro CPU with 18 GB of RAM. The total runtime is less than 5 minutes.

**Results.** The results are summarized in figure 1. Since ADDITIVE-PEGGING and MULTIPLICATIVE-PEGGING achieve almost the same competitive ratio and fairness for all instance types and values of $\varepsilon$ we only present ADDITIVE-PEGGING in figure 1 but include the code of both in the supplementary material. Our algorithms are consistently either the best or close to the best in terms of both competitive ratio and fairness for all different pairs of instance types and $\varepsilon$ values. Before discussing the results of each instance type individually it is instructive to mention some characteristics of our benchmarks. While DYNKIN does not use predictions and is therefore bound to suboptimal competitive ratios when predictions are accurate, we note that it accepts the maximum value candidate with probability at least $1/e$, i.e., it is $1/e$-fair. When predictions are non-informative this is an upper bound on the attainable fairness for any algorithm whether it uses predictions or not. HIGHEST-PREDICTION is expected to perform well when the highest prediction matches the true highest value candidate and poorly when the latter is not true. In Almost-constant for small values of $\varepsilon$ all candidates have very close true values and all algorithms except DYNKIN have a competitive ratio close to $1$. DYNKIN may not accept any candidate and this is why its performance is poorer than the rest of the algorithms. Note that as $\varepsilon$ increases both our algorithms perform significantly better than all other benchmarks.

In terms of fairness, predictions do not offer any information regarding the ordinal comparison between candidates' true values and this is why for small values of $\varepsilon$ the probability of HIGHEST-PREDICTION and LEARNED-DYNKIN of accepting the best candidate is close to $1/100 = 1/n$, i.e., random. Here, the fairness of our algorithms and DYNKIN is similar and close to $1/e$. In both Uniform and Adversarial we observe that for small values of $\varepsilon$ the highest predicted candidate is the

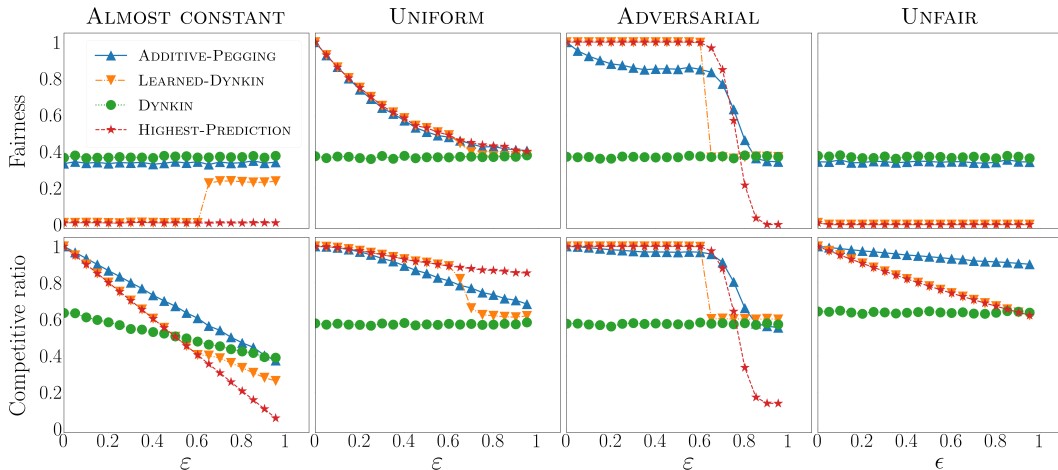

Figure 1: Competitive ratio and fairness of different algorithms, for each instance type and level of $\varepsilon$.

true highest and ADDITIVE-PEGGING, LEARNED-DYNKIN and HIGHEST-PREDICTION all accept that candidate having a very close performance both in terms of fairness and competitive ratio. For higher values of $\varepsilon$ the fairness of those algorithms deteriorates similarly and it approaches again $0.37 \simeq 1/e$. In Unfair our algorithms outperform all other benchmarks in terms of competitive ratio for all values of $\varepsilon$ and achieve a close to optimal fairness. This is expected as our algorithms are particularly suited for cases where predictions may be accurate but unfair.

Overall, our algorithms are the best-performing and most robust. The HIGHEST-PREDICTION algorithm does perform slightly better on Uniform instances and Adversarial instances under most values of $\varepsilon$, but performs consistently worse on Almost-constant and Unfair instances, especially in terms of fairness. Our algorithms perform better than LEARNED-DYNKIN in almost all situations.

## 6 Limitations and Future Work

We study a notion of fairness that is tailored to the secretary problem with predictions and build our algorithms based on this notion. However, there are alternative notions of fairness one could consider in applications such as hiring, as well as variations of the secretary problem that capture other features in these applications. While our model allows for arbitrary bias in the predictions we assume that the true value of a candidate is fully discovered upon arrival, and define fairness based on hiring the best candidate (who has the highest true value) with a reasonable probability. Thus, we ignore considerations such as bias in how we get the true value of a candidate (e.g., via an interview process). In addition, as noted in Section 1, we use an *individual fairness* notion which does not model other natural desiderata like hiring from underprivileged populations or balance the hiring probabilities across different populations. These are considerations with potentially high societal impact which our algorithms do not consider and are interesting directions for future work on fair selection with predictions.

Regarding trade-offs in our guarantees: for the single-secretary problem, we can improve the fairness guarantee from $1/16$ to $0.074$ by optimizing the constants in our algorithm. However, we choose not to do so, as the performance increase is marginal, and we aim to keep the proof as simple as possible. Additionally, as we noted in Section 1.1 any constant $C$ for smoothness implies an upper bound of $F = 1/3 + o(1)$ for fairness. Finding the Pareto-optimal curve in terms of smoothness and fairness is an interesting direction. The main challenge in achieving a smooth trade-off between fairness and smoothness is as follows: any bound on $C$ for smoothness implies a competitive ratio of $1 - C\epsilon$, which reaches a ratio of 1 when the predictions are exactly correct. Thus, regardless of the smoothness guarantee, we must achieve a competitive ratio of 1 when predictions are fully accurate. This constraint makes it challenging to improve the fairness guarantee $F$, even at the cost of a less favorable smoothness constant $C$.

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

# A Additional discussion and missing analysis for single secretary

## A.1 Unfair outcomes in previous work

In this section we present the learning-augmented algorithms of [2] and [30], and argue that they fail to satisfy simultaneously the smoothness and fairness desiderata described in section 2. We follow the same notation as in the main paper where the $i^*, \hat{\imath}$ denote the index of the candidate with maximum true and predicted value respectively. Since the algorithm in [2] requires only the prediction about the maximum value but not the identity of that candidate, we use the symbol $\hat{u}^*$ to denote such value.

---

**Algorithm 3** LEARNED-DYNKIN [30]

---

$\theta \leftarrow 0.646$, $t \leftarrow 0.313$, mode $\leftarrow$ Prediction
**while** agent $i$ arrives at time $t_i$ **do**
    $\tau \leftarrow \max_{j \prec i} u_j$
    **if** $|1 - \hat{u}_i / u_i| > \theta$ **then**
        mode $\leftarrow$ Secretary.
    **if** mode = Prediction and $i = \hat{\imath}$ **then**
        $\mathcal{A} \leftarrow i$.
    **if** mode = Secretary and $t_i > t$ and $u_i > \tau$ **then**
        $\mathcal{A} \leftarrow i$.

---

---

**Algorithm 4** VALUE-MAXIMIZATION SECRETARY [2]

---

**Input:** parameters $c, \lambda$ such that $\lambda \geq 0$ and $c \geq 1$.
$t^* \leftarrow \exp\{W^{-1}(-1/(ce))\}$, $t^{**} \leftarrow \exp\{W^0(-1/(ce))\}$
**while** agent $i$ arrives at time $t_i$ **do**
    $\tau^* \leftarrow \max_{j:t_j < t^*} u_j$
    **if** $t^* < t_i < t^{**}$ **and** $u_i > \max\{\tau^*, \hat{u}^* - \lambda\}$ **then**
        $\mathcal{A} \leftarrow i$
    $\tau^{**} \leftarrow \max_{j:t_j < t^{**}} u_j$
    **if** $t_i \geq t^{**}$ **and** $u_i > \tau^{**}$ **then**
        $\mathcal{A} \leftarrow i$

---

LEARNED-DYNKIN of Fujii and Yoshida [30] receives a predicted valuation for all candidates and defines the prediction error of candidate $i$ as $|1 - \hat{u}_i / u_i|$. If the prediction error of a candidate is higher than $\theta$ then it switches to Secretary mode where it mimics the classical DYNKIN algorithm where all candidates are rejected for a constant fraction of the stream and after that rejection phase the first candidate whose valuation is the maximum overall is hired. Note that if all candidates have very low prediction error then LEARNED-DYNKIN remains in the Prediction mode and the candidate with the higher prediction is hired. One instance where LEARNED-DYNKIN never accepts candidate $i^*$ is the following: there are two candidates with true valuations $u_1 = 1 + \theta'/2, u_2 = 1$ and predicted valuations $\hat{u}_1 = 1 + \theta'/2, \hat{u}_2 = 1 + \theta'$ where $\theta' < \theta = 0.646$. The prediction error is equal to $\theta' < \theta = 0.646$. Consequently LEARNED-DYNKIN does not switch to Prediction mode, and it never accepts the candidate with true valuation $1 + \theta'/2$, violating the Fairness desideratum.

VALUE-MAXIMIZATION SECRETARY of [2] receives only one prediction regarding the maximum value $\hat{u}^*$ and the prediction error is defined as $\varepsilon = |u_{i^*} - \hat{u}^*|$. The latter algorithm is parametrized by $\lambda \geq 0$ and $c \geq 1$ which control the relationship between the robustness and smoothness bounds. VALUE-MAXIMIZATION SECRETARY has three distinct phases defined by the time ranges $[0, t^*], (t^*, t^{**})$ and $[t^{**}, 1]$ respectively, where $t^*, t^{**}$ are defined using the Lambert functions $W^{-1}$ and $W^0$. The first phase is used as an "exploration" phase where all candidates are rejected and at the end of the phase a threshold $\tau^*$ is computed. During the second phase the algorithm accepts a candidate if and only if the true value of the candidate is larger than $\max\{\tau^*, \hat{u}^* - \lambda\}$. Finally, if at the end of the second phase no candidate has been selected then the algorithm accepts a candidate if their true value is the maximum so far (note that in the pseudocode this is done by computing the threshold $\tau^{**}$).

To demonstrate the failure of VALUE-MAXIMIZATION SECRETARY , let $\mathcal{A}$ be the random variable denoting the candidate it accepts. For any $\varepsilon \in (0, 1)$ we define an instance with predicted maximum

value $\hat{u}^* = 1 - \varepsilon$ and true values $\{u_1, u_2, \ldots, u_n\}$ where all numbers are distinct, $u_1 = 1$ and $u_i \in [0, \varepsilon]$, $\forall i \in \{2, \ldots, n\}$. The prediction error of that instance is $\varepsilon$. Note that if $i^* = 1$ arrives in the first phase then the maximum value of a candidate that VALUE-MAXIMIZATION SECRETARY can accept in the second and third phases is at most $\varepsilon$. Thus, we can upper bound the expected value of candidate $\mathcal{A}$ as follows: $\mathbf{E}[u_{\mathcal{A}}] \leq P[t_{i^*} \geq t^*] \cdot u_{i^*} + P[t_{i^*} < t^*] \cdot \varepsilon = (1 - t^*)u_{i^*} + t^* \varepsilon$.

We emphasize that in the learning-augmented setting, there is no assumption regarding the quality of the prediction; thus, the parameters $c$ and $\lambda$ cannot depend on the prediction error. For any parameter $c \geq 1$ (and any $\lambda$), we have that $t^* > 0$ is a constant that is bounded away from $0$. Towards a contradiction, assume that VALUE-MAXIMIZATION SECRETARY satisfies the smoothness desideratum described in section 2 for some parameter $C$. Then, we have that $\mathbf{E}[u_{\mathcal{A}}] \geq u_{i^*} - C \cdot \varepsilon$. Consequently, $(1 - t^*)u_{i^*} + t^* \varepsilon \geq u_{i^*} - C \cdot \varepsilon \xrightarrow{u_{i^*}=1} (1 - t^*) + t^* \varepsilon \geq 1 - C \cdot \varepsilon$ which leads to contradiction when we let $\varepsilon \to 0$.

## A.2    The PEGGING algorithm

In this subsection, we generalize the ADDITIVE-PEGGING algorithm so as to provide fair and smooth algorithms for different prediction error definitions. We use the direct sum symbol $\oplus$ to denote the addition or multiplication operation, $\ominus$ for its inverse operation, i.e., subtraction or division, and $\mathbf{0}$ for the identity element of $\oplus$, i.e, either the number $0$ if $\oplus$ denotes the classical addition operation or $1$ if $\oplus$ denotes the multiplication operation.

To model the "difference" between predictions and true values we use a function $\epsilon : \Re_{>0} \times \Re_{>0} \to \Re_{>0}$. We assume that $\epsilon$ receives as input only tuples of strictly positive reals so that the multiplicative error definition of [30] is not ill-defined[3]. Furthermore, as noted in the main paper, we assume distinct true and predicted values. This is achieved by introducing arbitrarily small perturbations, a simplification that does not affect our algorithm's performance and simplifies proofs by avoiding potentially cumbersome case distinctions arising from subtractions involving zero.

Our techniques require the following inequality to be true:

**Assumption 6.** $u \oplus (\mathbf{0} + \epsilon(u, \hat{u})) \geq \hat{u} \geq u \oplus (\mathbf{0} - \epsilon(u, \hat{u})), \forall u, \hat{u} \in \Re_{>0}, u \neq \hat{u}$

Assumption 6 essentially demands that a prediction $\hat{u}$ can be upper and lower bounded as a function of the true value $u$ and the "distance" $\epsilon(u, \hat{u})$ between true and predicted value. While the notation is left abstract to highlight the generality of Theorem 7, we provide examples on how to instantiate the error function and operators for natural prediction errors such as the absolute difference that we used in ADDITIVE-PEGGING and a multiplicative prediction error function that is used by Fujii and Yoshida [30]. To be specific, it is easy to check that Assumption 6 holds when:

- $\oplus$ is addition, $\mathbf{0} = 0$, and $\epsilon(u, \hat{u}) = |\hat{u} - u| = |u - \hat{u}|$;
- $\oplus$ is multiplication, $\mathbf{0} = 1$, and $\epsilon(u, \hat{u}) = |1 - \hat{u}/u|$;
- $\oplus$ is multiplication, $\mathbf{0} = 1$, and $\epsilon(u, \hat{u}) = |1 - \max\{u, \hat{u}\}/\min\{u, \hat{u}\}|$ (this follows from above because $|1 - \max\{u, \hat{u}\}/\min\{u, \hat{u}\}| \geq |1 - \hat{u}/u|$).

It is worth noting that Assumption 6 does not hold when $\oplus$ represents the scalar multiplication, $\mathbf{0} = 1$, and $\epsilon(u, \hat{u}) = |1 - u/\hat{u}|$. However, this definition of $\epsilon(u, \hat{u})$ is arguably less conventional than $\epsilon(u, \hat{u}) = |1 - \hat{u}/u|$ which is employed in [30].

Using the new abstract notation, we write $\epsilon_i = \epsilon(u_i, \hat{u}_i)$ and define the prediction error of our instance as $\varepsilon = \max_i \epsilon_i$. We can generalize ADDITIVE-PEGGING to PEGGING by making the following modifications:

1. Update the condition $u_i > \hat{u}_{\hat{\imath}} - \varepsilon_{t_i} \Leftrightarrow u_i + \varepsilon_{t_i} > \hat{u}_{\hat{\imath}}$ to $u_i \oplus (\mathbf{0} + \varepsilon_{t_i}) > \hat{u}_{\hat{\imath}}$.

2. Redefine the update rule of the "pegging" set from $I^{\text{pegged}} \leftarrow \{j \succ \hat{\imath} : u_{\hat{\imath}} < \hat{u}_j + \varepsilon_{t_{\hat{\imath}}}\}$ to $I^{\text{pegged}} \leftarrow \{j \succ \hat{\imath} : u_{\hat{\imath}} \oplus (\mathbf{0} - \varepsilon_{t_{\hat{\imath}}}) < \hat{u}_j\}$.

We proceed stating and proving the main theorem of this subsection.

---

[3]While we could allow zero-valued inputs by extending the image set to $\mathbb{R}_{>0} \cup \{\infty\}$, we avoid this for simplicity.

**Algorithm 5** PEGGING

//* The algorithm stops when it accepts a candidate by executing $\mathcal{A} \leftarrow i$. *//
**Initialization:** $I^{\text{pegged}} \leftarrow \emptyset$
**while** agent $i$ arrives at time $t_i$ **do**
    **if** $i \in I^{\text{pegged}}$ **then**
        **if** $|I^{\text{pegged}}| = 1$ **then**
            $\mathcal{A} \leftarrow i$
        **else**
            $I^{\text{pegged}} \leftarrow I^{\text{pegged}} \setminus \{i\}$
    $\mathcal{F} \leftarrow (u_i > \max_{j \prec i} u_j) \wedge (t_i > 1/2), \mathcal{C} \leftarrow (i = \hat{\imath}), \varepsilon_{t_i} \leftarrow \max_{j : t_j \leq t_i} \epsilon_j$
    **if** $\mathcal{C} \wedge \mathcal{F}$ **then**
        $\mathcal{A} \leftarrow i$
    **else if** $\overline{\mathcal{C}} \wedge \mathcal{F}$ **then**
        **if** $u_i \oplus (\mathbf{0} + \varepsilon_{t_i}) > \hat{u}_{\hat{\imath}}$ **then**
            $\mathcal{A} \leftarrow i$
    **else if** $\mathcal{C} \wedge \overline{\mathcal{F}}$ **then**
        $I^{\text{pegged}} \leftarrow \{j \succ \hat{\imath} : u_{\hat{\imath}} \oplus (\mathbf{0} - \varepsilon_{t_i}) < \hat{u}_j\}$
        **if** $I^{\text{pegged}} = \emptyset$ **then**
            $\mathcal{A} \leftarrow i$

**Theorem 7.** *Suppose that $\oplus$ represents either scalar addition or scalar multiplication, with $\ominus$ being its inverse. Suppose that the error function $\epsilon(\cdot, \cdot)$ satisfies Assumption 6. Then* PEGGING *accepts the maximum value candidate with probability at least $\frac{1}{16}$ and its expected value is at least* $\max\{u_{i^*} \oplus (\mathbf{0} - \varepsilon) \oplus (\mathbf{0} - \varepsilon) \ominus (\mathbf{0} + \varepsilon) \ominus (\mathbf{0} + \varepsilon), \frac{1}{16} u_{i^*}\}$

*Proof.* The proof that PEGGING always accepts a candidate is the same as in Lemma 1 and therefore we omit it. To prove the smoothness bound we argue that the selected candidate $i$ has a true value that is close to the true value of $\hat{\imath}$ which is in turn close to the true value of $i^*$.

To lower bound $\hat{u}_{\hat{\imath}}$ we use the right-hand side of Assumption 6 and the fact that $\hat{\imath}$ is the candidate with the maximum predicted value, and get $\hat{u}_{\hat{\imath}} \geq \hat{u}_{i^*} \geq u_{i^*} \oplus (\mathbf{0} - \epsilon_{i^*})$.

We also lower bound $u_{\hat{\imath}}$ as follows: from the left-hand side of Assumption 6 we get that $u_{\hat{\imath}} \oplus (\mathbf{0} + \epsilon_{\hat{\imath}}) \geq \hat{u}_{\hat{\imath}} \Rightarrow u_{\hat{\imath}} \geq \hat{u}_{\hat{\imath}} \ominus (\mathbf{0} + \epsilon_{\hat{\imath}})$. Combining the latter with the previously argued lower bound $\hat{u}_{\hat{\imath}} \geq u_{i^*} \oplus (\mathbf{0} - \epsilon_{i^*})$ we have:

$$u_{\hat{\imath}} \geq \hat{u}_{\hat{\imath}} \ominus (\mathbf{0} + \epsilon_{\hat{\imath}})$$
$$\geq u_{i^*} \oplus (\mathbf{0} - \epsilon_{i^*}) \ominus (\mathbf{0} + \epsilon_{\hat{\imath}})$$

The latter inequality lower bounds the value accrued by our algorithm whenever candidate $\hat{\imath}$ is selected, i.e., literal $C$ is true at the time of the selection. If a candidate is accepted using the if-statement corresponding to literal $\overline{\mathcal{C}} \wedge \mathcal{F}$ then we have $u_i \oplus (\mathbf{0} + \varepsilon_{t_i}) \geq \hat{u}_{\hat{\imath}} \Rightarrow u_i \geq \hat{u}_{\hat{\imath}} \ominus (\mathbf{0} + \varepsilon_{t_i})$ and again using the lower bound on $\hat{u}_{\hat{\imath}}$ we get

$$u_i \geq u_{i^*} \oplus (\mathbf{0} - \epsilon_{i^*}) \ominus (\mathbf{0} + \varepsilon_{t_i})$$

Note that up until this moment we considered all but one case where our algorithm may accept a candidate and terminate. Indeed, we still need to lower bound the value of the last candidate in the pegging set since it may be the one accepted by our algorithm. Let the index of that candidate be $i^{pegged}$.

Note that for all $j \in I^{\text{pegged}}$, it holds that $u_{\hat{\imath}} \oplus (\mathbf{0} - \varepsilon_{t_{\hat{\imath}}}) < \hat{u}_j$. Thus for $i^{pegged}$, using the left hand side of Assumption 6 we have:

$$u_{i^{\text{pegged}}} \oplus (\mathbf{0} + \epsilon_{i^{\text{pegged}}}) \geq \hat{u}_{i^{\text{pegged}}} \Rightarrow$$

$$u_{i^{\text{pegged}}} \geq \hat{u}_{i^{\text{pegged}}} \ominus (\mathbf{0} + \epsilon_{i^{\text{pegged}}}) \xrightarrow{i^{\text{pegged}} \in I^{\text{pegged}}}$$

$$u_{i^{\text{pegged}}} \geq u_{\hat{\imath}} \oplus (\mathbf{0} - \varepsilon_{t_{\hat{\imath}}}) \ominus (\mathbf{0} + \epsilon_{i^{\text{pegged}}}) \xrightarrow{\text{lower bound of } u_{\hat{\imath}}}$$

$$u_{i^{\text{pegged}}} \geq u_{i^*} \oplus (\mathbf{0} - \epsilon_{i^*}) \ominus (\mathbf{0} + \epsilon_{\hat{\imath}}) \oplus (\mathbf{0} - \varepsilon_{t_{\hat{\imath}}}) \ominus (\mathbf{0} + \epsilon_{i^{\text{pegged}}}) \Rightarrow$$

$$u_{i^{\text{pegged}}} \geq u_{i^*} \oplus (\mathbf{0} - \varepsilon) \oplus (\mathbf{0} - \varepsilon) \ominus (\mathbf{0} + \varepsilon) \ominus (\mathbf{0} + \varepsilon)$$

Combining all the lower bounds on the value of the accepted candidate we deduce the first part of the lower bound.

We proceed proving fairness and, consequently robustness. The proof of the latter follows the same line of thinking as the proof of Lemma 2. We repeat it here to conform with the abstract notation and also assume (as in Lemma 2) that the number of candidates is at least 3 (if there are two candidates, then the proof remains essentially the same and the fairness bound improves to $1/4$). Let $\tilde{\imath}$ be the index of the candidate with the highest true value except $i^*$ and $\hat{\imath}$, i.e., $\tilde{\imath} = \text{argmax}_{i \neq i^*, \hat{\imath}} u_i$.

If $i^* = \hat{\imath}$ we define event $E = \{t_{\tilde{\imath}} < 1/2 < t_{i^*}\}$ for which $P[E] = 1/4$. $E$ implies that our algorithm does not accept any candidate until time $t_{i^*}$. Indeed, note that at any point in time before $t_{i^*}$, both literals $\mathcal{F}$ and $\mathcal{C}$ are simultaneously false. On the contrary, at time $t_{i^*}$, both $\mathcal{C}$ and $\mathcal{F}$ are true and our algorithm accepts $i^*$.

On the other hand, if $i^* \neq \hat{\imath}$ we distinguish between two sub-cases, namely when $\{u_{\hat{\imath}} \oplus (\mathbf{0} - \epsilon_{\hat{\imath}}) < \hat{u}_{i^*}\}$ and $\{u_{i^*} \oplus (\mathbf{0} + \epsilon_{i^*}) > \hat{u}_{\hat{\imath}}\}$. We continue arguing that $\{u_{\hat{\imath}} \oplus (\mathbf{0} - \epsilon_{\hat{\imath}}) < \hat{u}_{i^*}\} \vee \{u_{i^*} \oplus (\mathbf{0} + \epsilon_{i^*}) > \hat{u}_{\hat{\imath}}\}$ is always true. Indeed if $\{u_{\hat{\imath}} \oplus (\mathbf{0} - \epsilon_{i^*}) \geq \hat{u}_{i^*}\}$ we have that:

$$u_{\hat{\imath}} \oplus (\mathbf{0} - \epsilon_{\hat{\imath}}) \geq \hat{u}_{i^*} \xrightarrow{\text{right hand side of } Assumption\ 6}$$

$$u_{\hat{\imath}} \oplus (\mathbf{0} - \epsilon_{\hat{\imath}}) \geq u_{i^*} \oplus (\mathbf{0} - \epsilon_{i^*}) \xrightarrow{(\star)}$$

$$u_{i^*} \oplus (\mathbf{0} + \epsilon_{i^*}) \geq u_{\hat{\imath}} \oplus (\mathbf{0} + \epsilon_{\hat{\imath}}) \xrightarrow{\text{left hand side of } Assumption\ 6}$$

$$u_{i^*} \oplus (\mathbf{0} + \epsilon_{i^*}) \geq \hat{u}_{\hat{\imath}}$$

where for $(\star)$ we need to prove that

$$u_{\hat{\imath}} \oplus (\mathbf{0} - \epsilon_{\hat{\imath}}) \geq u_{i^*} \oplus (\mathbf{0} - \epsilon_{i^*}) \Rightarrow u_{i^*} \oplus (\mathbf{0} + \epsilon_{i^*}) \geq u_{\hat{\imath}} \oplus (\mathbf{0} + \epsilon_{\hat{\imath}})$$

If $\oplus$ denotes the addition operation then we have that:

$$u_{\hat{\imath}} + (0 - \epsilon_{\hat{\imath}}) \geq u_{i^*} + (0 - \epsilon_{i^*}) \Leftrightarrow$$

$$\epsilon_{i^*} - \epsilon_{\hat{\imath}} \geq u_{i^*} - u_{\hat{\imath}} \xrightarrow{u_{i^*} > u_{\hat{\imath}}}$$

$$\epsilon_{i^*} > \epsilon_{\hat{\imath}} \xrightarrow{u_{i^*} > u_{\hat{\imath}}}$$

$$u_{i^*} + (0 + \epsilon_{i^*}) > u_{\hat{\imath}} + (0 + \epsilon_{\hat{\imath}})$$

Likewise if $\oplus$ denotes the multiplication operation then:

$$u_{\hat{\imath}} \cdot (1 - \epsilon_{\hat{\imath}}) \geq u_{i^*} \cdot (1 - \epsilon_{i^*}) \Leftrightarrow$$

$$u_{i^*} \epsilon_{i^*} - u_{\hat{\imath}} \epsilon_{\hat{\imath}} \geq u_{i^*} - u_{\hat{\imath}} \xrightarrow{u_{i^*} > u_{\hat{\imath}}}$$

$$u_{i^*} \epsilon_{i^*} > u_{\hat{\imath}} \epsilon_{\hat{\imath}} \xrightarrow{u_{i^*} > u_{\hat{\imath}}}$$

$$u_{i^*} \epsilon_{i^*} + u_{i^*} > u_{\hat{\imath}} \epsilon_{\hat{\imath}} + u_{\hat{\imath}} \Leftrightarrow$$

$$u_{i^*} \cdot (1 + \epsilon_{i^*}) > u_{\hat{\imath}} \cdot (1 + \epsilon_{\hat{\imath}})$$

We now define two events $E_1$ and $E_2$ which imply that $i^*$ is always accepted whenever $\{u_{\hat{\imath}} \oplus (\mathbf{0} - \epsilon_{\hat{\imath}}) < \hat{u}_{i^*}\}$ and $\{u_{i^*} \oplus (\mathbf{0} + \epsilon_{i^*}) > \hat{u}_{\hat{\imath}}\}$ are true respectively.

If $\{u_{\hat{\imath}} \oplus (\mathbf{0} - \epsilon_{\hat{\imath}}) < \hat{u}_{i^*}\}$, then we define event $E_1 = \{t_{\tilde{\imath}} < 1/2\} \wedge \{t_{\hat{\imath}} < 1/2\} \wedge \{1/2 < t_{i^*}\}$ which is composed by 3 independent events and it happens with probability $P[E_1] = 1/2^3 = 1/8$. $E_1$ implies that $t_{\hat{\imath}} < t_{i^*} \Rightarrow \varepsilon_{t_{i^*}} \geq \epsilon_{\hat{\imath}}$, thus we can deduce that:

$$u_{i^*} \oplus (\mathbf{0} + \varepsilon_{t_{i^*}}) \overset{u_{i^*} > u_{\hat{\imath}}, \varepsilon_{t_{i^*}} \geq \epsilon_{\hat{\imath}}}{\geq} u_{\hat{\imath}} \oplus (\mathbf{0} + \epsilon_{\hat{\imath}}) \overset{\text{left hand side of } Assumption\ 6}{\geq} \hat{u}_{\hat{\imath}}$$

Consequently, if until time $t_{i^*}$ all candidates are rejected, $E_1$ implies that $\overline{\mathcal{C}} \wedge \mathcal{F} \wedge \{u_{i^*} \oplus (\mathbf{0} + \varepsilon_{t_{i^*}}) \geq \hat{u}_{\hat{\imath}}\}$ is true at time $t_{i^*}$ and candidate $i^*$ is hired. To argue that no candidate is accepted before time $t_{i^*}$, note that $\mathcal{F}$ is false at all times before $t_{i^*}$ and at time $t_{\hat{\imath}}$ (when literal $\mathcal{C}$ is true) the set $\{j \succ \hat{\imath} : u_{\hat{\imath}} \oplus (\mathbf{0} - \varepsilon_{t_{\hat{\imath}}}) < \hat{u}_j\} \supseteq \{j \succ \hat{\imath} : u_{\hat{\imath}} \oplus (\mathbf{0} - \epsilon_{\hat{\imath}}) < \hat{u}_j\}$ contains $i^*$.

If $\{u_{i^*} \oplus (\mathbf{0} + \epsilon_{i^*}) > \hat{u}_{\hat{\imath}}\}$, then we define $E_2 = \{t_{\tilde{\imath}} < 1/2 < t_{i^*} < t_{\hat{\imath}}\}$ which happens with probability

$$\begin{aligned}
P[E_2] &= P[t_{\tilde{\imath}} < 1/2] \cdot P[1/2 < t_{i^*} < t_{\hat{\imath}}] \\
&= P[t_{\tilde{\imath}} < 1/2] \cdot P[1/2 < \min\{t_{i^*}, t_{\hat{\imath}}\} \wedge \min\{t_{i^*}, t_{\hat{\imath}}\} = t_{i^*}] \\
&= P[t_{\tilde{\imath}} < 1/2] \cdot P[1/2 < \min\{t_{i^*}, t_{\hat{\imath}}\}] \cdot P[\min\{t_{i^*}, t_{\hat{\imath}}\} = t_{i^*}] \\
&= (1/2) \cdot (1/4) \cdot (1/2) = 1/16
\end{aligned}$$

Note that until time $t_{i^*}$ no candidate is accepted since $\mathcal{C}$ and $\mathcal{F}$ are both false at all times. Indeed, between times 0 and $1/2$ only $\hat{\imath}$ could have been accepted but its arrival time is after $t_{i^*}$, and between times $1/2$ and $t_{i^*}$ no candidate has a true value larger than $u_{\tilde{\imath}}$. Finally, note that at time $t_{i^*}$ we have $\varepsilon_{t_{i^*}} \geq \epsilon_{i^*}$ and consequently $\overline{\mathcal{C}} \wedge \mathcal{F} \wedge \{(\mathbf{0} + \varepsilon_{t_{i^*}}) \oplus u_{i^*} > \hat{u}_{\hat{\imath}}\}$ is true and $i^*$ gets accepted. $\qquad\square$

Note that by instantiating $\oplus, \ominus$ to the usual scalar addition and subtraction and defining $\epsilon(u, \hat{u}) = |u - \hat{u}|$ we get that, $\varepsilon = \max_i |u_i - \hat{u}_i|$ and Theorem 7 recovers Theorem 3.

We further demonstrate the generality of PEGGING by instantiating $\oplus, \ominus, \epsilon$ and recovering similar smoothness and robustness bounds as [30] while also ensuring fairness. Fujii and Yoshida [30] define the prediction error as $\varepsilon = \max_i |1 - \hat{u}_i/u_i|$ and design an algorithm which accepts a candidate $i$ whose expected value is at least $u_{i^*} \max\{(1 - \varepsilon)/(1 + \varepsilon), 0.215\}$. Since $(1 - \varepsilon)/(1 + \varepsilon) \geq 1 - 2\varepsilon$ the latter algorithm satisfies the Smoothness desideratum of Section 2, but, as we prove in Appendix A.1 it violates the Fairness desideratum.

To that end, let MULTIPLICATIVE-PEGGING be the instantiation of PEGGING when $\oplus, \ominus$ denote the classical multiplication and division operations, and $\epsilon(u, \hat{u}) = |1 - \hat{u}/u|$. We now use Theorem 7 to recover Theorem 4, which is restated below for convenience.

**Theorem 4.** *Let* $\varepsilon(\mathcal{I}) = \max_i |1 - \hat{u}_i/u_i|$ *and assume* $u_i, \hat{u}_i > 0 \; \forall i \in [n]$. *Then* MULTIPLICATIVE-PEGGING *satisfies fairness with* $F = 1/16$ *and selects a candidate* $\mathcal{A}$ *such that* $u_{\mathcal{A}} \geq u_{i^*} \cdot (1 - 4 \cdot \varepsilon(\mathcal{I}))$ *with probability 1.*

*Proof.* Function $\epsilon(u, \hat{u}) = |1 - \hat{u}/u|$ satisfies the properties of Assumption 6. Consequently, using Theorem 7 MULTIPLICATIVE-PEGGING accepts a candidate whose expected value is at least $u_{i^*} \max\{(1 - \varepsilon)^2/(1 + \varepsilon)^2, 1/16\} \geq u_{i^*} \max\{1 - 4\varepsilon, 1/16\}$, where we used the inequalities $1/(1 + \varepsilon) \geq (1 - \varepsilon)$ and $(1 - \varepsilon)^4 \geq 1 - 4\varepsilon$. $\qquad\square$

## B  Missing analysis for the k-secretary pegging algorithm

In Lemma 8 we prove that $k$-PEGGING satisfies the smoothness desideratum.

**Lemma 8.** $\sum_{j \in S} u_j \geq \sum_{i=1}^k u_{r_i} - 4k \, \varepsilon(\mathcal{I})$ , $\forall \mathcal{I}$ *with probability* 1.

*Proof.* Similarly to the single choice secretary problem we proceed in two steps, first we prove that

$$\sum_{i \in [k]} u_{r_i} - \sum_{i \in [k]} u_i \leq 2k \, \varepsilon$$

and then we prove that

$$\sum_{i \in [k]} u_i - \sum_{j \in S} u_j \leq 2k \, \varepsilon$$

Note that combining those two inequalities is enough to prove the current lemma.

The first inequality is proven as follows:

$$\sum_{i=1}^{k} u_{r_i} \leq_{(1)} \sum_{i=1}^{k} (\hat{u}_{r_i} + \varepsilon) \leq_{(2)} k\,\varepsilon + \sum_{i=1}^{k} \hat{u}_i \leq_{(3)} k\,\varepsilon + \sum_{i=1}^{k} (u_i + \varepsilon) = 2k\,\varepsilon + \sum_{i=1}^{k} u_i$$

where (1) and (3) are by definition of $\varepsilon$ and (2) since $\hat{u}_i$ is $i^{th}$ largest predicted value.

We proceed to argue that:

$$\sum_{i \in [k]} u_i - \sum_{j \in S} u_j \leq 2k\,\varepsilon$$

We now define an injective function $\mathrm{m} : [k] \to S$ for which we have:

$$u_i - u_{\mathrm{m}(i)} \leq 2\,\varepsilon, \ \forall i \in [k]$$

Note that the existence of such a function implies the desired $\sum_{i \in [k]} u_i - \sum_{j \in H} u_j \leq 2k\,\varepsilon$ inequality.

Note that each candidate $j' \in [k]$ is initially added to $H$. During the execution of our algorithm candidate $j'$ may be either (1) deleted from $H$ without being added to $B$ or (2) added to $B$.

The first case where $j'$ is deleted from $H$ without being added to $B$, occurs either in case $2, 3a$, or $4b$ of the algorithm. Let $i$ be the current candidate at the time $t_i$ when the latter happens. If *case 2 or 3a* happens then $j' = i$, we define $\mathrm{m}(j') = j'$ and $u_{j'} - u_{\mathrm{m}(j')} = 0 \leq 2\,\varepsilon$. If *case 4b* happens then we have that at that time $t_i$, $j' \in \{j \in H : u_i > \hat{u}_j - \varepsilon_{t_i}\}$ and we define $\mathrm{m}(i) = j'$. Consequently, we conclude that $u_i - u_{\mathrm{m}(i)} = u_i - u_{j'} \leq u_i - \hat{u}_{j'} + \varepsilon_{t_{j'}} \leq \epsilon_i + \varepsilon_{t_{j'}} \leq 2\,\varepsilon$.

We now consider the cases where $j'$ is added to $B$ during the execution of our algorithm. Note that for that to happen $j'$ must be added to $B$ at time $t_{j'}$ via *case 3b*. In that case, candidate $\mathrm{peg}\,(j')$ either remains in $P$ until time $t_{\mathrm{peg}(j')}$ and it is added to $S$ at that time or it is deleted from $P$ earlier. In both cases, $j'$ is removed from $B$ at the respective time. Thus, we conclude that $j'$ gets deleted from $B$ at time $t_{\mathrm{peg}(j')}$ or before. If the deletion happens at time $t_{\mathrm{peg}(j')}$ then it must happen through *case 1*, we define $\mathrm{m}(j') = \mathrm{peg}\,(j')$ and we have that $u_{j'} - u_{\mathrm{m}(j')} = u_{j'} - u_{\mathrm{peg}(j')} \leq \hat{u}_{\mathrm{peg}(j')} + \varepsilon_{t_{j'}} - u_{\mathrm{peg}(j')} = (\hat{u}_{\mathrm{peg}(j')} - u_{\mathrm{peg}(j')}) + \varepsilon_{t_{j'}} \leq \epsilon_{\mathrm{peg}(j')} + \varepsilon_{t_{j'}} \leq 2\,\varepsilon$, where in the first inequality we used that $\hat{u}_{\mathrm{peg}(j')} > u_{j'} - \varepsilon_{t_{j'}}$ since $\mathrm{peg}\,(j')$ was pegged by $j'$ at time $t_{j'}$. If $j'$ is deleted from $B$ before time $t_{\mathrm{peg}(j')}$ it must happen via *subcase 4a* due to the arrival of a candidate $l$ that is added to $S$. In that case we define $\mathrm{m}(j') = l$ and from the condition of *subcase 4a* we have that $j \in E$ at time $t_{j'}$ we have that $u_{j'} - u_{\mathrm{m}(j')} = u_{j'} - u_l < 0 \leq 2\,\varepsilon$. $\qquad\square$

We now move on to prove the fairness desideratum.

**Lemma 9.** *For all $i \in [k]$: $P[r_i \in S] \geq (1/3)^{k+5}$*

*Proof.* We start arguing that if $r_i \in [k]$, i.e., if candidate $r_i$ is among the $k$-highest prediction candidates, then $P[r_i \in S] \geq (1/2)^{k+1}$. We define event $\mathcal{E}$, which happens with probability $(1/2)^{k+1}$ and implies that $r_i$ is added to the solution $S$. Let

$$\mathcal{E} = \bigwedge_{l \in [k+1] \setminus \{i\}} \{t_{r_l} < 1/2\} \wedge \{t_{r_i} > 1/2\}$$

During the interval $[0, 1/2]$, only candidates in $[k]$ may be added to $S$. In addition, $\mathcal{E}$ implies that the threshold $\tau$ is greater than $u_{r_{k+1}}$ after time $1/2$, and only candidate $r_i$ has a value exceeding $\tau$ after that time. Thus, other than $r_i$ the only candidates that may be added to $S$ during the interval $[1/2, t_{r_i}]$, are candidates pegged by a candidate $j \in [k]$. In summary, $\mathcal{E}$ implies that $S$ consists of candidates in $[k]$ or candidates which were pegged by some candidate in $j \in [k]$ which is not part of the solution set $S$. Thus, $|S| < k$ before time $t_{r_i}$, and at that time $r_i$ is added to $S$ through *case 1*.

In the rest of the proof, we focus on the case where $r_i \notin [k]$. Note that since $i \in [k]$ and $r_i \notin [k]$ then $\exists j \in [k]$ such that $u_{r_i} \geq u_{r_k} > u_j$, i.e., $j$ has a true value which is not among the $k$-highest true values. We now argue that $\{u_j < \hat{u}_{r_i} + \epsilon_j\} \vee \{u_{r_i} > \hat{u}_j - \epsilon_{r_i}\}$ is always true. Similarly to the proof

of Lemma 2, assume towards a contradiction that both inequalities can be inverted and hold at the same time, then we end up in a contradiction as follows:

$$u_j \geq \hat{u}_{r_i} + \epsilon_j \xrightarrow{u_{r_i} > u_j} u_{r_i} > \hat{u}_{r_i} + \epsilon_j \Rightarrow u_{r_i} - \hat{u}_{r_i} > \epsilon_j \xrightarrow{\epsilon_{r_i} \geq u_{r_i} - \hat{u}_{r_i}} \epsilon_{r_i} > \epsilon_j$$

$$u_{r_i} \leq \hat{u}_j - \epsilon_{r_i} \xrightarrow{u_j < u_{r_i}} u_j < \hat{u}_j - \epsilon_{r_i} \Rightarrow \epsilon_{r_i} < \hat{u}_j - u_j \xrightarrow{\epsilon_j \geq \hat{u}_j - u_j} \epsilon_{r_i} < \epsilon_j$$

For each of those cases, i.e., whether $\{u_j < \hat{u}_{r_i} + \epsilon_j\}$ or $\{u_{r_i} > \hat{u}_j - \epsilon_{r_i}\}$ is true we define an event which implies that $r_i$ is added to the solution set $S$.

If $\{u_{r_i} > \hat{u}_j - \epsilon_{r_i}\}$ is true then we define the following event:

$$\mathcal{E} = \bigwedge_{\{r_1,\dots,r_{k+2}\}\setminus\{r_i,j\}} \{t_l < 1/2\} \wedge \{1/2 < t_{r_i} < t_j\}$$

Note that:

$$P[\mathcal{E}] = P[1/2 < t_{r_i} < t_j] \cdot \prod_{\{r_1,\dots,r_{k+2}\}\setminus\{r_i,j\}} P[t_{r_l} < 1/2]$$

$$= P[1/2 < \min\{t_{r_i}, t_j\} \wedge \{t_{r_i} < t_j\}] \cdot \prod_{\{r_1,\dots,r_{k+2}\}\setminus\{r_i,j\}} (1/2)$$

$$= P[1/2 < \min\{t_{r_i}, t_j\}] \cdot P[t_{r_i} < t_j] \cdot \prod_{\{r_1,\dots,r_{k+2}\}\setminus\{r_i,j\}} (1/2)$$

$$\geq P[1/2 < \min\{t_{r_i}, t_j\}] \cdot P[t_{r_i} < t_j] \cdot (1/2)^{k+2}$$

$$= (1/4) \cdot (1/2) \cdot (1/2)^{k+2}$$

$$= (1/2)^{k+5}$$

We now argue that $\mathcal{E}$ implies that $r_i$ is added to $S$.

The first literal of $\mathcal{E}$ ensures that $\tau \geq u_{r_{k+1}}$ after time $1/2$ and, consequently, the only candidate with a true value higher than the threshold $\tau$ at any time $t \in [1/2, t_{r_i}]$ is $r_i$. That observation implies that conditions of *case 4* are true only for $r_i$. We now argue that: (1) before time $t_{r_i}$ less than $k$ candidates are added to $S$; and (2) the conditions of *subcase 4b* are true for $r_i$ at time $t_{r_i}$.

For (1) first note that: (a) initially we have $|S| = |B| = 0$, $|H| = k$; (b) at all times our algorithm maintains the invariant $B \cap H = \emptyset$, $B \cup H \subseteq [k]$; and (c) every time a candidate is added to the solution $S$ then a candidate is deleted from either $B$, as in *case 1* and *subcase 4a*, or $H$ as in *case 2*, *subcase3a*, and *subcase 4b*. Thus, at all times $|B| + |H| + |S|$ remains constant and since initially is equal to $k$ we conclude that at all times $|B| + |H| + |S| = k$. In addition, a candidate not yet arrived may be removed from $H$ only through *subcase 4b*. Since we argued that conditions of *case 4* are true only for $r_i$, we have that right before $r_i$'s arrival $j \in H$ and $|S| = k - |B| - |H| \leq k - |H| \leq k - 1 < k$. For (2), to argue that conditions of *subcase 4b* are met at time $t_{r_i}$, it is enough to prove that $j \in \{j' : u_{r_i} > \hat{u}_{j'} - \varepsilon_{t_{r_i}}\}$ (note that $j \in H$ and $\mathcal{E}$ implies that $t_j > t_{r_i}$). To see this, note that by the definition of $\varepsilon_{t_{r_i}}$ it holds that $\varepsilon_{t_{r_i}} \geq \epsilon_{r_i}$ and consequently, $\{j' : u_{r_i} > \hat{u}_{j'} - \varepsilon_{t_{r_i}}\} \supseteq \{j' : u_{r_i} > \hat{u}_{j'} - \epsilon_{r_i}\} \ni j$.

Before proceeding to the second case we introduce the following notation:

$$t_{\text{peg}(j)} = \begin{cases} t_l & \text{if } \exists l : l = \text{peg}(j) \\ \infty & \text{otherwise} \end{cases}$$

that is, if at time $t_j$ candidate $l$ is added to the pegging set $P$ then we use $t_{\text{peg}(j)}$ to denote the arrival time of candidate $l$. However, if at time $t_j$ no candidate is added to set $P$ then we define $t_{\text{peg}(j)}$ to be equal to $\infty$ so that the literal $\{t_{\text{peg}(j)} > x\}$ is true for every $x \in \Re$.

We now analyze the case where $\{u_j < \hat{u}_{r_i} + \epsilon_j\}$ is true and define the following event:

$$\mathcal{E} = \bigwedge_{\{r_1,\dots,r_{k+1}\}\setminus\{r_i,j\}} \{t_{r_l} < 1/3\} \wedge \{1/3 < t_j < 1/2\} \wedge \{t_{r_i} > 1/2\} \wedge \{t_{\text{peg}(j)} \geq t_{r_i}\}$$

To simplify notation, let $P_j$ be the random variable denoting the pegging set at time $t_j$ before the execution of the while loop because of $j$'s arrival. We let $F_j = \{j' \succ j : u_j < \hat{u}_{j'} + \varepsilon_{t_j}\} \setminus (P_j \cup [k])$ be the random variable which contains all candidates that could be "pegged" at time $t_j$.

In addition we define event $\mathcal{T}$ as follows:
$$\mathcal{T} = \bigwedge_{\{r_1,\ldots,r_{k+1}\}\setminus\{r_i,j\}} \{t_{r_l} < 1/3\} \wedge \{1/3 < t_j < 1/2\} \wedge \{t_{r_i} > 1/2\}$$

Before lower bounding the probability of event $\mathcal{E}$ we argue that:
$$P\big[t_{\text{peg}(j)} \geq t_{r_i} \mid \mathcal{T}\big] \geq 2/3$$

Let $\mathcal{F}_j$ denote the set of all non-empty subsets of $[n]$ such that $P[F_j = f_j \mid \mathcal{T}] > 0$. Note that $P[F_j = \emptyset \mid \mathcal{T}] + \sum_{f_j \in \mathcal{F}_j} P[F_j = f_j \mid \mathcal{T}] = 1$.

From the law of total probability we have:

$$P\big[t_{\text{peg}(j)} \geq t_{r_i} \mid \mathcal{T}\big] \tag{1}$$
$$= P\big[\{t_{\text{peg}(j)} \geq t_{r_i}\} \wedge \{F_j = \emptyset\} \mid \mathcal{T}\big] + P\big[\{t_{\text{peg}(j)} \geq t_{r_i}\} \wedge \{F_j \neq \emptyset\} \mid \mathcal{T}\big] \tag{2}$$
$$= P[F_j = \emptyset \mid \mathcal{T}] + P\big[\{t_{\text{peg}(j)} \geq t_{r_i}\} \wedge \{F_j \neq \emptyset\} \mid \mathcal{T}\big] \tag{3}$$
$$= P[F_j = \emptyset \mid \mathcal{T}] + \sum_{f_j \in \mathcal{F}_j} P\big[\{t_{\text{peg}(j)} \geq t_{r_i}\} \wedge \{F_j = f_j\} \mid \mathcal{T}\big] \tag{4}$$
$$= P[F_j = \emptyset \mid \mathcal{T}] + \sum_{f_j \in \mathcal{F}_j} P\big[t_{\text{peg}(j)} \geq t_{r_i} \mid \{F_j = f_j\} \wedge \mathcal{T}\big] \cdot P[F_j = f_j \mid \mathcal{T}] \tag{5}$$

Where from (2) to (3) we use that if $F_j = \emptyset$ then the condition of *subcase3b* is false, thus no candidate is "pegged" and consequently $t_{\text{peg}(j)} = \infty$. From (3) to (4) we used that $\{F_j \neq \emptyset\} = \bigvee_{f_j \in \mathcal{F}_j}\{F_j = f_j\}$. We now focus on lower bounding the summation term.

$$\sum_{f_j \in \mathcal{F}_j} P\big[t_{\text{peg}(j)} \geq t_{r_i} \mid \{F_j = f_j\} \wedge \mathcal{T}\big] \cdot P[F_j = f_j \mid \mathcal{T}] =$$
$$\sum_{f_j \in \mathcal{F}_j : \text{peg}(j)=r_i} P\big[t_{\text{peg}(j)} \geq t_{r_i} \mid \{F_j = f_j\} \wedge \mathcal{T}\big] \cdot P[F_j = f_j \mid \mathcal{T}]+$$
$$+ \sum_{f_j \in \mathcal{F}_j : \text{peg}(j)\neq r_i} P\big[t_{\text{peg}(j)} \geq t_{r_i} \mid \{F_j = f_j\} \wedge \mathcal{T}\big] \cdot P[F_j = f_j \mid \mathcal{T}] =$$
$$\sum_{f_j \in \mathcal{F}_j : \text{peg}(j)=r_i} 1 \cdot P[F_j = f_j \mid \mathcal{T}] + \sum_{f_j \in \mathcal{F}_j : \text{peg}(j)\neq r_i} P\big[t_{\text{peg}(j)} \geq t_{r_i} \mid \{F_j = f_j\} \wedge \mathcal{T}\big] \cdot P[F_j = f_j \mid \mathcal{T}]$$

We proceed lower bounding the term $P\big[t_{\text{peg}(j)} \geq t_{r_i} \mid \{F_j = f_j\} \wedge \mathcal{T}\big]$ for all $f_j \in \mathcal{F}_j : \text{peg}(j) \neq r_i$. Note that the conditioning $\{F_j = f_j\} \wedge \mathcal{T}$ changes the distribution of random variables $t_{\text{peg}(j)}, t_{r_i}$ as follows: $t_{r_i}$ is uniformly drawn from $[1/2, 1]$ and $t_{\text{peg}(j)}$ is uniformly drawn from $[z, 1]$ for some $z \in [1/3, 1/2]$ which equals the realization of the random variable $t_j$. We define a random variable $\tilde{t}_{\text{peg}(j)}$ which is stochastically dominated by $t_{\text{peg}(j)}$ and is drawn uniformly from $[1/3, 1]$ as follows: let $\tilde{t}$ be uniformly drawn from $[1/3, z]$ and $B \sim Bernoulli((z - 1/3)/(1/2 - 1/3))$ then we define:
$$\tilde{t}_{\text{peg}(j)} = B \cdot \tilde{t} + (1 - B) \cdot t_{\text{peg}(j)}$$

Note that since $\tilde{t} \leq t_{\text{peg}(j)}$ then also $\tilde{t}_{\text{peg}(j)} \leq t_{\text{peg}(j)}$ holds almost surely.

Therefore we have:
$$P\big[t_{\text{peg}(j)} \geq t_{r_i} \mid \{F_j = f_j\} \wedge \mathcal{T}\big] \geq P\big[\tilde{t}_{\text{peg}(j)} \geq t_{r_i} \mid \{F_j = f_j\} \wedge \mathcal{T}\big]$$
$$\geq 3/8$$

We proceed to bound the initial summation as follows:

$$\sum_{f_j \in \mathcal{F}_j} P\Big[t_{l_{f_j}} \geq t_{r_i} \mid \{F_j = f_j\} \wedge \mathcal{T}\Big] \cdot P[F_j = f_j \mid \mathcal{T}] =$$

$$= \sum_{f_j \in \mathcal{F}_j : l_{f_j} = r_i} 1 \cdot P[F_j = f_j \mid \mathcal{T}] + \sum_{f_j \in \mathcal{F}_j : l_{f_j} \neq r_i} P\Big[t_{l_{f_j}} \geq t_{r_i} \mid \{F_j = f_j\} \wedge \mathcal{T}\Big] \cdot P[F_j = f_j \mid \mathcal{T}]$$

$$\geq \sum_{f_j \in \mathcal{F}_j : l_{f_j} = r_i} 1 \cdot P[F_j = f_j \mid \mathcal{T}] + \sum_{f_j \in \mathcal{F}_j : l_{f_j} \neq r_i} (3/8) \cdot P[F_j = f_j \mid \mathcal{T}]$$

$$\geq (3/8) \sum_{f_j \in \mathcal{F}_j} P[F_j = f_j \mid \mathcal{T}]$$

We then have:

$$P\big[t_{\mathrm{peg}(j)} \geq t_{r_i} \mid \mathcal{T}\big] \tag{6}$$

$$= P[F_j = \emptyset \mid \mathcal{T}] + \sum_{f_j \in \mathcal{F}_j} P\Big[t_{l_{f_j}} \geq t_{r_i} \mid \{F_j = f_j\} \wedge \mathcal{T}\Big] \cdot P[F_j = f_j \mid \mathcal{T}] \tag{7}$$

$$\geq P[F_j = \emptyset \mid \mathcal{T}] + (3/8) \sum_{f_j \in \mathcal{F}_j} P[F_j = f_j \mid \mathcal{T}] \tag{8}$$

$$\geq (3/8) \cdot \left( P[F_j = \emptyset \mid \mathcal{T}] + \sum_{f_j \in \mathcal{F}_j} P[F_j = f_j \mid \mathcal{T}] \right) \tag{9}$$

$$= 3/8 \tag{10}$$

We are now ready to lower bound the probability of event $\mathcal{E}$ as follows

$$\begin{aligned}
P[\mathcal{E}] &= P[\mathcal{T}] \cdot P\big[t_{\mathrm{peg}(j)} \geq t_{r_i} \mid \mathcal{T}\big] \\
&\geq P[\mathcal{T}] \cdot (3/8) \\
&= \prod_{\{r_1, \dots, r_{k+1}\} \setminus \{r_i, j\}} P[t_l < 1/3] \cdot P[1/3 < t_j < 1/2] \cdot (3/8) \\
&\geq (1/3)^k \cdot (1/2 - 1/3) \cdot (3/8) \\
&= (1/3)^{k+3}
\end{aligned}$$

Similar to the analysis of the first case the first literal of $\mathcal{E}$ ensures that the only candidate which may be accepted after time $1/2$ without being at any point in time in the pegging set $P$ is $r_i$. In addition, since $t_j < 1/2$ then we have that $j$ remains in $H$ until at least time $t_j$. Indeed, a candidate in $H$ that has not arrived yet may be removed from set $H$ only through *case 4b* which happens exclusively after time $1/2$. We now analyze $\mathcal{E}$'s implications regarding the execution of our algorithm at $j$'s arrival by distinguishing between two mutually exclusive cases, that is whether $r_i$ is in $P$ before time $t_j$ or not.

If $r_i$ is not in the pegging set exactly before time $t_j$ then we have that the conditions of *case 3b* are true. Indeed note that since $\epsilon_j \leq \varepsilon_{t_j}$ we have:

$$\{j' \succ j : u_j < \hat{u}_{j'} + \varepsilon_{t_j}\} \setminus (P \cup [k]) \supseteq \{j' \succ j : u_j < \hat{u}_{j'} + \epsilon_j\} \setminus (P \cup [k]) \ni r_i$$

Thus, we are in the case where at time $t_j$ a candidate (which may be $r_i$) is added to the pegging set, candidate $j$ is added to $B$ and $t_{\mathrm{peg}(j)} < \infty$. Due to the literal $t_{\mathrm{peg}(j)} \geq t_{r_i}$ of $\mathcal{E}$ and the fact that the only candidate which may be accepted after time $1/2$ without being at any point in time in the pegging set $P$ is $r_i$, we can deduce that at time $t_{r_i}$ $j$ is still in $B$. Thus, at time $t_{r_i}$ the conditions of *subcase 4a* are true and $r_i$ is added to $S$. If $r_i$ is in the pegging set exactly before time $t_j$ then since the conditions of *case 4* are false for any candidate except possibly $r_i$ we can deduce that at time $t_{r_i}$, candidate $r_i$ is still in the pegging set $P$ and is added to the solution through *case 1*.

Combining all the different lower bounds on $P[r_i \in S]$ we conclude the lemma. $\qquad\square$

**Lemma 10.** *For all $i \in \{1, 2, \ldots, k\}$: $P[r_i \in S] \geq \frac{1 - \frac{i+13}{k}}{256}$*

*Proof.* Let $\delta' > 12/k$ be such that $i = (1 - \delta')k - 1$. We now argue that proving $P[r_i \in S] \geq \delta'/256$ suffices to prove the lemma.

First, we underline that such a $\delta'$ exists only for $i < k - 13$. Indeed,

$$i = (1 - \delta')k - 1 \Rightarrow \delta' = 1 - \frac{i+1}{k} \xrightarrow{\delta' > 12/k} i < k - 13$$

For all $i < k - 13$ we have:

$$P[r_i \in S] \geq \delta'/256 > \frac{1 - \frac{i+1}{k}}{256} > \frac{1 - \frac{i+13}{k}}{256}$$

For $i \geq k - 13$ the statement of the lemma is vacuous, since:

$$P[r_i \in S] \geq 0 \geq \frac{1 - \frac{i+13}{k}}{256}$$

Consequently, from now on we focus on proving that $P[r_i \in S] \geq \delta'/256$. We do so by defining an event $\mathcal{E}$ for which $P[\mathcal{E}] \geq \delta'/256$ and argue that $\mathcal{E}$ implies $r_i$ being accepted.

Before defining $\mathcal{E}$ we need to introduce some auxiliary notation. We call replacement set and denote by $R$ the set of indexes initially in $H$ with value lower than $u_{r_i}$, i.e., $R = \{j : u_{r_i} > u_j\} \cap [k]$ and by $j^{worse} = \operatorname{argmax}_{j \in R} \epsilon_j$ the index of the candidate with the highest error in $R$. For any $t, t' \in [0, 1]$ we define the random variable $A_{t,t'} = \{j : t \leq t_j \leq t'\} \setminus \{r_i, j^{worse}\}$ which contains all indexes except $r_i$ and $j^{worse}$ of candidates arrived between times $t$ and $t'$. Also, for $x \in [n]$ we define the set function $L_x : 2^{[n]} \to 2^{[n]}$, such that for any subset $Y \subseteq [n]$, $L_x(Y)$ contains the $x$ indexes with highest true value in $Y$. For $\delta \in (0, 1/2)$ let (a) $R_1 = R \cap A_{0,1/2+\delta}$ and $R_2 = R \cap A_{1/2+\delta,1}$ be the random variables denoting all candidates of $R \setminus \{j^{worse}\}$ arriving before and after time $1/2 + \delta$ respectively; and (b) let $M = L_{\lfloor(1+4\delta)k\rfloor}(A_{0,1/2+\delta}) \cap A_{1/2,1/2+\delta}$ denote the random variable containing candidates which arrived between times $1/2$ and $1/2 + \delta$ with the $\lfloor(1 + 4\delta)k\rfloor$ higher true value among the ones arrived before time $1/2 + \delta$ (excluding $r_i$ and $j^{worse}$).

We now define event $\mathcal{E}$. Let $\delta = \delta'/16$:

$$\mathcal{E} = \left\{|R_2| \geq \frac{1/2 - 2\delta}{1 - \delta}\delta'k\right\} \wedge \{|M| < 4\delta k\} \wedge \{1/2 < t_{r_i} < 1/2 + \delta\} \wedge \{t_{j^{worse}} < t_{r_i}\}$$

The literal $\{|M| < 4\delta k\}$ implies that at most $4\delta k$ candidates not in $H$ or not "pegged" by a previously arrived candidate in $H$ may be added to our solution between times $1/2$ and $1/2+\delta$. In addition, each candidate in $M$ through *subcase 4b* may delete from $H$ at most one candidate with arrival time after $1/2 + \delta$. Consequently, the number of candidates in $R$ that are in $H$ until time $1/2 + \delta$ are at least $|R_2| - |M|$. We first argue that $\{|R_2| \geq \frac{1/2-2\delta}{1-\delta}\delta'k\}$ and $|M| < 4\delta k$ implies that $|R_2| - |M| > 1$, i.e., until time $1/2 + \delta$ at least one candidate from the replacement set $R$ is still in $H$ (note that initially we have $R \subseteq H = [k]$).

$$\begin{aligned}
|R_2| - |M| &> \frac{1/2 - 2\delta}{1 - \delta}\delta'k - 4\delta k \\
&> \frac{1}{3}\delta'k - 4\delta k \\
&> (\delta'/3 - 4\delta)k \\
&\geq (\delta'/3 - 4\delta'/16)k \\
&\geq (\delta'/12)k \\
&> 1
\end{aligned}$$

where in the second inequality we used that $\delta = \delta'/16 \leq 1/16 < 1/10$.

We continue arguing that $\mathcal{E}$ implies that the conditions of *subcase 4b* are true at time $t_{r_i}$. For every candidate $j \in R_2$ it holds that:

$$u_{r_i} > u_j \geq \hat{u}_j - \epsilon_j \geq \hat{u}_j - \epsilon_{j^{worse}} \geq \hat{u}_j - \varepsilon_{t_{r_i}}$$

Where the first inequality comes from the definition of set $R$ since $R_2 \subseteq R$, the second from the definition of the error, the third from the definition of $j^{worse}$ and the last from the fact that $t_{j^{worse}} < t_{r_i}$ implies $\varepsilon_{t_{r_i}} \geq \epsilon_{j^{worse}}$. Thus, we have that at time $t_{r_i}$ there is at least one candidate $j$ in $H$ for which $u_{r_i} \geq \hat{u}_j - \varepsilon_{t_{r_i}}$, consequently the conditions of *subcase 4a* are true, and candidate $r_i$ is added to our solution.

We now lower bound the probability of event $\mathcal{E}$. Since $R_2$ and $M$ do not contain neither $r_i$ nor $j^{worse}$ we have that events $\{|R_2| \geq \frac{1/2-2\delta}{1-\delta}\delta'k\}$, $\{|M| < 4\delta k\}$ are independent of the time arrival of $j^{worse}$ and $r_i$. Thus,

$$
\begin{aligned}
P[\mathcal{E}] &= P\left[\{|R_2| \geq \frac{1/2-2\delta}{1-\delta}\delta'k\} \wedge \{|M| < 4\delta k\}\right] \cdot P[1/2 < t_{r_i} < 1/2 + \delta] \cdot P[t_{j^{worse}} < 1/2] \\
&= P\left[\{|R_2| \geq \frac{1/2-2\delta}{1-\delta}\delta'k\} \wedge \{|M| < 4\delta k\}\right] \cdot \delta \cdot (1/2) \\
&= (\delta'/32) \cdot P\left[\{|R_2| \geq \frac{1/2-2\delta}{1-\delta}\delta'k\} \wedge \{|M| < 4\delta k\}\right] \\
&= (\delta'/32) \cdot P\left[\{|R_1| < |R| - \frac{1/2-2\delta}{1-\delta}\delta'k\} \wedge \{|M| < 4\delta k\}\right]
\end{aligned}
$$

We continue by lower bounding the second term of the last expression. We do so by defining a random variable $\mathcal{M}$ which is independent of $R_1$ and it is such that $\mathcal{M}$ stochastically dominates $|M|$. We prove the stochastic dominance of $\mathcal{M}$ using a coupling argument.

Note that every candidate $i$ accepts an arrival time $t_i$ uniformly at random from $[0,1]$. We now describe an equivalent procedure to create the arrival times $t_i$. Each candidate $i$ draws three independent random variables $B_i \sim Bernoulli(1/2 + \delta)$, $t_i^1 \sim Uniform([0, 1/2 + \delta])$ and $t_i^2 \sim Uniform([1/2 + \delta, 1])$. Note that we can construct random variables $t_i$ using $B_i, t_i^1$ and $t_i^2$ as follows:

$$
t_i = B_i \cdot t_i^1 + (1 - B_i) \cdot t_i^2
$$

Let $l = |L_{(1+4\delta)k}(A_{0,1/2+\delta})|$ and denote by $h_1, \ldots, h_l$ the set of candidates in $L_{(1+4\delta)k}(A_{0,1/2+\delta})$. We define random variables $\tilde{t}_1, \ldots, \tilde{t}_{(1+4\delta)k}$ as follows: For each $j \in \{1, \ldots, l\}$ we define $\tilde{t}_j = t_{h_j}^1$ and for each $j \in \{l+1, \ldots, 1 + 4\delta k\}$ we define $\tilde{t}_j \sim Uniform([0, 1/2 + \delta])$. We define $\mathcal{M} = \sum_{i=1}^{\lfloor(1+4\delta)k\rfloor} \mathbb{I}\{\tilde{t}_j > 1/2\}$ and since $\mathcal{M} - |M| = \sum_{i=l+1}^{\lfloor(1+4\delta)k\rfloor} \mathbb{I}\{\tilde{t}_j > 1/2\}$ we have that $\mathcal{M} \geq |M|$ almost surely. Note that random variables $\tilde{t}_j$ and random variables $B_i$ are independent. Consequently, since $|R_1| = \sum_{i \in R} B_i$ we have that events $\{\mathcal{M} < y\}$ and $\{|R_1| < x\}$ are independent. Combining these observations we have:

$$
\begin{aligned}
&P\left[\{|R_1| < |R| - \frac{1/2-2\delta}{1-\delta}\delta'k\} \wedge \{|M| < 4\delta k\}\right] \\
\geq &P\left[\{|R_1| < |R| - \frac{1/2-2\delta}{1-\delta}\delta'k\} \wedge \{\mathcal{M} < 4\delta k\}\right] \\
\geq &P\left[|R_1| < |R| - \frac{1/2-2\delta}{1-\delta}\delta'k\right] \cdot P[\mathcal{M} < 4\delta k] \\
\geq &P\left[|R_2| \geq \frac{1/2-2\delta}{1-\delta}\delta'k\right] \cdot P[\mathcal{M} < 4\delta k]
\end{aligned}
$$

To upper bound $P[\mathcal{M} < 4\delta k]$ note that $\mathbf{E}[\mathcal{M}] = \lfloor(1+4\delta)k\rfloor \cdot \frac{\delta}{1/2+\delta} < 3\delta k$, where the last inequality holds since $1/2 > 1/16 \geq \delta'/16 = \delta$. From Markov's inequality, we have that:

$$P[\mathcal{M} > 4\delta k] = P[\mathcal{M} > (4/3) \cdot 3\delta k]$$
$$\leq P[\mathcal{M} > (4/3) \cdot \mathbf{E}[\mathcal{M}]]$$
$$\leq (3/4)$$

Consequently

$$P[\mathcal{M} < 4\delta k] > (1/4)$$

Note that since $i < (1 - \delta')k - 1$ we have that $|R| \geq \delta'k + 1$.

$$\mathbf{E}[|R_2|] = |R \setminus \{j^{worse}\}|(1 - 1/2 - \delta) \geq \delta'k(1/2 - \delta)$$

From Markov's inequality, we have that:

$$P\left[|R_2| > \frac{1/2 - 2\delta}{1 - \delta}\delta'k\right] \geq P\left[|R_2| > \frac{1/2 - 2\delta}{(1 - \delta) \cdot (1/2 - \delta)} \cdot \mathbf{E}[|R_2|]\right]$$
$$\geq \frac{(1 - \delta) \cdot (1/2 - \delta)}{1/2 - 2\delta}$$
$$= \frac{(1 - \delta) \cdot (1 - 2\delta)}{1 - 4\delta}$$
$$\geq (1 - \delta)$$

Consequently,

$$P[\mathcal{E}] \geq (\delta'/32) \cdot (1/4) \cdot (1 - \delta)$$
$$\geq (\delta'/32) \cdot (1/4) \cdot (1/2)$$
$$\geq (\delta'/256)$$

$\square$

]

**Theorem 5.** $k$-PEGGING *satisfies smoothness and fairness for $k$-secretary with $C = 4k$ and* $F_\ell = \max\left\{(1/3)^{k+5}, \frac{1 - (\ell + 13)/k}{256}\right\}$ *for all $\ell = 1, \ldots, k$.*

*Proof.* The theorem follows directly from Lemmas 8 to 10. $\square$

## C   Additional experimental results

In this section we present the competitive ratio achieved by each algorithm as a function of the parameter $\varepsilon$ which controls the predictions error in each of the datasets defined in Section 5.

Table 1: Competitive ratio (mean ± std deviation) on Almost-constant

| $\varepsilon$ | ADD.-PEGGING | MULT.-PEGGING | LEARNED-DYNKIN | DYNKIN | HIGHEST-PRED. |
|---|---|---|---|---|---|
| 0.0 | 1.00 ± 0.00 | 1.00 ± 0.00 | 1.00 ± 0.00 | 0.64 ± 0.48 | 1.00 ± 0.00 |
| 0.05 | 0.97 ± 0.02 | 0.97 ± 0.02 | 0.95 ± 0.00 | 0.62 ± 0.47 | 0.95 ± 0.00 |
| 0.1 | 0.93 ± 0.05 | 0.93 ± 0.05 | 0.90 ± 0.01 | 0.61 ± 0.46 | 0.90 ± 0.01 |
| 0.15 | 0.90 ± 0.07 | 0.90 ± 0.07 | 0.85 ± 0.01 | 0.61 ± 0.45 | 0.85 ± 0.01 |
| 0.2 | 0.87 ± 0.09 | 0.87 ± 0.09 | 0.80 ± 0.02 | 0.59 ± 0.45 | 0.80 ± 0.02 |
| 0.25 | 0.83 ± 0.12 | 0.83 ± 0.12 | 0.75 ± 0.02 | 0.57 ± 0.44 | 0.75 ± 0.02 |
| 0.3 | 0.80 ± 0.14 | 0.80 ± 0.14 | 0.70 ± 0.03 | 0.55 ± 0.44 | 0.70 ± 0.03 |
| 0.35 | 0.77 ± 0.17 | 0.77 ± 0.17 | 0.65 ± 0.03 | 0.54 ± 0.43 | 0.65 ± 0.03 |
| 0.4 | 0.74 ± 0.19 | 0.74 ± 0.19 | 0.60 ± 0.04 | 0.53 ± 0.43 | 0.60 ± 0.04 |
| 0.45 | 0.70 ± 0.21 | 0.70 ± 0.21 | 0.55 ± 0.04 | 0.53 ± 0.43 | 0.55 ± 0.04 |
| 0.5 | 0.67 ± 0.24 | 0.67 ± 0.24 | 0.50 ± 0.05 | 0.51 ± 0.43 | 0.50 ± 0.05 |
| 0.55 | 0.64 ± 0.26 | 0.64 ± 0.26 | 0.46 ± 0.06 | 0.49 ± 0.43 | 0.46 ± 0.06 |
| 0.6 | 0.61 ± 0.29 | 0.61 ± 0.29 | 0.41 ± 0.06 | 0.48 ± 0.43 | 0.41 ± 0.06 |
| 0.65 | 0.57 ± 0.31 | 0.57 ± 0.31 | 0.40 ± 0.36 | 0.46 ± 0.43 | 0.36 ± 0.06 |
| 0.7 | 0.54 ± 0.33 | 0.54 ± 0.33 | 0.38 ± 0.36 | 0.45 ± 0.44 | 0.31 ± 0.07 |
| 0.75 | 0.51 ± 0.36 | 0.51 ± 0.36 | 0.36 ± 0.37 | 0.44 ± 0.44 | 0.26 ± 0.07 |
| 0.8 | 0.47 ± 0.38 | 0.47 ± 0.38 | 0.33 ± 0.38 | 0.42 ± 0.45 | 0.21 ± 0.08 |
| 0.85 | 0.44 ± 0.40 | 0.44 ± 0.40 | 0.31 ± 0.39 | 0.41 ± 0.46 | 0.16 ± 0.08 |
| 0.9 | 0.40 ± 0.42 | 0.40 ± 0.42 | 0.28 ± 0.40 | 0.39 ± 0.46 | 0.11 ± 0.09 |
| 0.95 | 0.37 ± 0.45 | 0.37 ± 0.45 | 0.26 ± 0.41 | 0.38 ± 0.47 | 0.06 ± 0.10 |

Table 2: Competitive ratio (mean ± std deviation) on Uniform

| $\varepsilon$ | ADD.-PEGGING | MULT.-PEGGING | LEARNED-DYNKIN | DYNKIN | HIGHEST-PRED. |
|---|---|---|---|---|---|
| 0.0 | 1.00 ± 0.00 | 1.00 ± 0.00 | 1.00 ± 0.00 | 0.57 ± 0.45 | 1.00 ± 0.00 |
| 0.05 | 1.00 ± 0.01 | 1.00 ± 0.01 | 1.00 ± 0.01 | 0.58 ± 0.45 | 1.00 ± 0.01 |
| 0.1 | 0.99 ± 0.02 | 0.99 ± 0.02 | 0.99 ± 0.02 | 0.58 ± 0.45 | 0.99 ± 0.02 |
| 0.15 | 0.98 ± 0.04 | 0.98 ± 0.04 | 0.99 ± 0.04 | 0.57 ± 0.45 | 0.99 ± 0.04 |
| 0.2 | 0.97 ± 0.07 | 0.97 ± 0.06 | 0.98 ± 0.05 | 0.57 ± 0.45 | 0.98 ± 0.05 |
| 0.25 | 0.96 ± 0.09 | 0.96 ± 0.08 | 0.97 ± 0.07 | 0.57 ± 0.45 | 0.97 ± 0.07 |
| 0.3 | 0.93 ± 0.12 | 0.94 ± 0.11 | 0.95 ± 0.08 | 0.58 ± 0.45 | 0.95 ± 0.08 |
| 0.35 | 0.91 ± 0.14 | 0.92 ± 0.12 | 0.94 ± 0.10 | 0.57 ± 0.45 | 0.94 ± 0.10 |
| 0.4 | 0.89 ± 0.16 | 0.90 ± 0.15 | 0.93 ± 0.11 | 0.56 ± 0.45 | 0.93 ± 0.11 |
| 0.45 | 0.87 ± 0.18 | 0.89 ± 0.16 | 0.92 ± 0.12 | 0.58 ± 0.45 | 0.92 ± 0.12 |
| 0.5 | 0.85 ± 0.20 | 0.87 ± 0.17 | 0.91 ± 0.13 | 0.58 ± 0.45 | 0.91 ± 0.13 |
| 0.55 | 0.83 ± 0.22 | 0.86 ± 0.19 | 0.90 ± 0.14 | 0.57 ± 0.45 | 0.90 ± 0.14 |
| 0.6 | 0.81 ± 0.23 | 0.84 ± 0.20 | 0.89 ± 0.14 | 0.57 ± 0.45 | 0.89 ± 0.14 |
| 0.65 | 0.79 ± 0.25 | 0.83 ± 0.21 | 0.83 ± 0.27 | 0.58 ± 0.45 | 0.89 ± 0.15 |
| 0.7 | 0.77 ± 0.26 | 0.81 ± 0.22 | 0.66 ± 0.41 | 0.57 ± 0.45 | 0.88 ± 0.15 |
| 0.75 | 0.76 ± 0.28 | 0.80 ± 0.23 | 0.63 ± 0.42 | 0.57 ± 0.45 | 0.87 ± 0.16 |
| 0.8 | 0.73 ± 0.29 | 0.78 ± 0.24 | 0.61 ± 0.43 | 0.57 ± 0.45 | 0.87 ± 0.16 |
| 0.85 | 0.72 ± 0.30 | 0.77 ± 0.25 | 0.62 ± 0.43 | 0.58 ± 0.45 | 0.86 ± 0.16 |
| 0.9 | 0.70 ± 0.31 | 0.76 ± 0.25 | 0.61 ± 0.43 | 0.57 ± 0.45 | 0.86 ± 0.17 |
| 0.95 | 0.68 ± 0.33 | 0.75 ± 0.26 | 0.61 ± 0.43 | 0.57 ± 0.45 | 0.85 ± 0.17 |

Table 3: Competitive ratio (mean ± std deviation) on Adversarial

| $\varepsilon$ | ADD.-PEGGING | MULT.-PEGGING | LEARNED-DYNKIN | DYNKIN | HIGHEST-PRED. |
|---|---|---|---|---|---|
| 0.0 | 1.00 ± 0.00 | 1.00 ± 0.00 | 1.00 ± 0.00 | 0.57 ± 0.45 | 1.00 ± 0.00 |
| 0.05 | 1.00 ± 0.01 | 1.00 ± 0.01 | 1.00 ± 0.00 | 0.57 ± 0.45 | 1.00 ± 0.00 |
| 0.1 | 0.99 ± 0.03 | 0.99 ± 0.03 | 1.00 ± 0.00 | 0.57 ± 0.45 | 1.00 ± 0.00 |
| 0.15 | 0.99 ± 0.05 | 0.99 ± 0.05 | 1.00 ± 0.00 | 0.57 ± 0.45 | 1.00 ± 0.00 |
| 0.2 | 0.98 ± 0.06 | 0.98 ± 0.06 | 1.00 ± 0.00 | 0.57 ± 0.45 | 1.00 ± 0.00 |
| 0.25 | 0.98 ± 0.07 | 0.98 ± 0.07 | 1.00 ± 0.00 | 0.57 ± 0.45 | 1.00 ± 0.00 |
| 0.3 | 0.97 ± 0.08 | 0.97 ± 0.08 | 1.00 ± 0.00 | 0.58 ± 0.45 | 1.00 ± 0.00 |
| 0.35 | 0.97 ± 0.09 | 0.97 ± 0.09 | 1.00 ± 0.00 | 0.57 ± 0.45 | 1.00 ± 0.00 |
| 0.4 | 0.97 ± 0.09 | 0.97 ± 0.09 | 1.00 ± 0.00 | 0.58 ± 0.45 | 1.00 ± 0.00 |
| 0.45 | 0.97 ± 0.09 | 0.97 ± 0.09 | 1.00 ± 0.00 | 0.58 ± 0.45 | 1.00 ± 0.00 |
| 0.5 | 0.97 ± 0.09 | 0.97 ± 0.09 | 1.00 ± 0.00 | 0.57 ± 0.45 | 1.00 ± 0.00 |
| 0.55 | 0.97 ± 0.09 | 0.97 ± 0.09 | 1.00 ± 0.01 | 0.57 ± 0.45 | 1.00 ± 0.01 |
| 0.6 | 0.97 ± 0.10 | 0.97 ± 0.10 | 1.00 ± 0.05 | 0.57 ± 0.45 | 1.00 ± 0.05 |
| 0.65 | 0.96 ± 0.13 | 0.96 ± 0.13 | 0.60 ± 0.43 | 0.57 ± 0.45 | 0.98 ± 0.13 |
| 0.7 | 0.91 ± 0.22 | 0.92 ± 0.21 | 0.60 ± 0.44 | 0.57 ± 0.45 | 0.88 ± 0.29 |
| 0.75 | 0.80 ± 0.32 | 0.81 ± 0.32 | 0.60 ± 0.43 | 0.58 ± 0.45 | 0.64 ± 0.41 |
| 0.8 | 0.66 ± 0.38 | 0.67 ± 0.39 | 0.60 ± 0.43 | 0.58 ± 0.45 | 0.34 ± 0.35 |
| 0.85 | 0.57 ± 0.40 | 0.56 ± 0.42 | 0.59 ± 0.44 | 0.57 ± 0.45 | 0.17 ± 0.16 |
| 0.9 | 0.57 ± 0.40 | 0.52 ± 0.43 | 0.61 ± 0.43 | 0.58 ± 0.45 | 0.14 ± 0.04 |
| 0.95 | 0.56 ± 0.40 | 0.49 ± 0.43 | 0.60 ± 0.44 | 0.58 ± 0.45 | 0.14 ± 0.04 |

Table 4: Competitive ratio (mean ± std deviation) on Unfair

| $\varepsilon$ | ADD.-PEGGING | MULT.-PEGGING | LEARNED-DYNKIN | DYNKIN | HIGHEST-PRED. |
|---|---|---|---|---|---|
| 0.0 | 1.00 ± 0.00 | 1.00 ± 0.00 | 1.00 ± 0.00 | 0.63 ± 0.48 | 1.00 ± 0.00 |
| 0.05 | 0.99 ± 0.01 | 0.99 ± 0.01 | 0.98 ± 0.00 | 0.63 ± 0.48 | 0.98 ± 0.00 |
| 0.1 | 0.99 ± 0.02 | 0.99 ± 0.02 | 0.95 ± 0.00 | 0.64 ± 0.48 | 0.95 ± 0.00 |
| 0.15 | 0.98 ± 0.02 | 0.98 ± 0.02 | 0.93 ± 0.00 | 0.65 ± 0.48 | 0.93 ± 0.00 |
| 0.2 | 0.98 ± 0.03 | 0.98 ± 0.03 | 0.91 ± 0.00 | 0.64 ± 0.48 | 0.91 ± 0.00 |
| 0.25 | 0.97 ± 0.04 | 0.97 ± 0.04 | 0.88 ± 0.00 | 0.64 ± 0.48 | 0.88 ± 0.00 |
| 0.3 | 0.96 ± 0.04 | 0.96 ± 0.04 | 0.86 ± 0.00 | 0.64 ± 0.48 | 0.86 ± 0.00 |
| 0.35 | 0.96 ± 0.05 | 0.96 ± 0.05 | 0.84 ± 0.00 | 0.64 ± 0.48 | 0.84 ± 0.00 |
| 0.4 | 0.95 ± 0.06 | 0.95 ± 0.06 | 0.82 ± 0.00 | 0.64 ± 0.48 | 0.82 ± 0.00 |
| 0.45 | 0.95 ± 0.06 | 0.95 ± 0.06 | 0.80 ± 0.00 | 0.63 ± 0.48 | 0.80 ± 0.00 |
| 0.5 | 0.94 ± 0.07 | 0.94 ± 0.07 | 0.78 ± 0.00 | 0.64 ± 0.48 | 0.78 ± 0.00 |
| 0.55 | 0.94 ± 0.08 | 0.94 ± 0.08 | 0.76 ± 0.00 | 0.64 ± 0.48 | 0.76 ± 0.00 |
| 0.6 | 0.93 ± 0.08 | 0.93 ± 0.08 | 0.74 ± 0.00 | 0.64 ± 0.48 | 0.74 ± 0.00 |
| 0.65 | 0.93 ± 0.09 | 0.93 ± 0.09 | 0.73 ± 0.00 | 0.64 ± 0.48 | 0.73 ± 0.00 |
| 0.7 | 0.92 ± 0.10 | 0.92 ± 0.10 | 0.71 ± 0.00 | 0.64 ± 0.48 | 0.71 ± 0.00 |
| 0.75 | 0.92 ± 0.10 | 0.92 ± 0.10 | 0.69 ± 0.00 | 0.64 ± 0.48 | 0.69 ± 0.00 |
| 0.8 | 0.92 ± 0.11 | 0.92 ± 0.11 | 0.67 ± 0.00 | 0.64 ± 0.48 | 0.67 ± 0.00 |
| 0.85 | 0.91 ± 0.11 | 0.91 ± 0.11 | 0.66 ± 0.00 | 0.64 ± 0.48 | 0.66 ± 0.00 |
| 0.9 | 0.91 ± 0.12 | 0.91 ± 0.12 | 0.64 ± 0.00 | 0.64 ± 0.48 | 0.64 ± 0.00 |
| 0.95 | 0.90 ± 0.12 | 0.90 ± 0.12 | 0.62 ± 0.00 | 0.64 ± 0.48 | 0.62 ± 0.00 |

