# OpenReview forum: "Fair Secretaries with Unfair Predictions"
_NeurIPS.cc/2024/Conference — NeurIPS 2024 poster_

### Official Review · Reviewer_Le4z · 2024-07-11

**Soundness:** 3
**Presentation:** 4
**Contribution:** 3
**Rating:** 7
**Confidence:** 4

**Summary:**

This work studies algorithms with untrusted predictions for secretary problems, considering fairness. In this paper, an algorithm is deemed fair if it can accept the best candidate with at least a constant probability. This fairness definition implies that a good candidate deserves a fair chance. The paper first demonstrates that the SOTA learning-augmented algorithm is unfair due to potentially biased predictions, meaning it may accept the best candidate with zero probability. Subsequently, the paper proposes a new algorithm that takes biased predictions as input but ensures both fairness and smoothness (which captures the algorithm's competitive performance). The design and analysis of the algorithm are based on an interesting pegging idea and can be extended to the k-secretary problem. Finally, extensive experiments are conducted to compare the proposed algorithm with SOTA algorithms, showcasing its advantages in both competitive ratios and fairness.

**Strengths:**

- This paper addresses an important and timely topic by investigating algorithmic fairness in the presence of potentially biased machine-learned predictions. The authors effectively identify the potential unfairness introduced by biased predictions and propose a novel algorithm to resolve these fairness issues within the context of secretary problems with predictions.

- The pegging idea used for the design and analysis of the algorithm for the secretary problem with predictions is both interesting and effective. Additionally, this idea can be extended to the k-secretary problem, and the treatment of general prediction errors is also noteworthy.

- Extensive empirical tests clearly demonstrate the advantages of the proposed algorithm over SOTA learning-augmented algorithms in terms of both fairness and competitiveness.

**Weaknesses:**

- Although the motivation for the fairness definition in the paper is understandable and appreciated, the definition itself remains somewhat vague. If I understand correctly, the current fairness definition is more relevant when the best candidate is significantly better than the second-best candidate. In such cases, a fair algorithm should select the best candidate with a non-zero probability, implying that a good candidate should have a better chance of being chosen. However, when the values of the top two candidates are very close, it is unclear how the algorithm ensures fairness without providing guarantees for the second-best candidate. The paper would benefit from a more comprehensive discussion and validation of the fairness definition.



- It is quantitatively unclear what the price of enforcing fairness in algorithms with predictions is. The proposed algorithm guarantees both smoothness C (that indicates the effectiveness of using the predictions) and fairness F. There should inherently be a trade-off between C and F, which is not discussed in the paper. And one natural question is whether the current algorithm is flexible and can be tuned to adjust the trade-off.

**Questions:**

- Can you provide formal comments on the definition of fairness? Is the current algorithm fair for the second-best candidate, especially when the value of the second-best candidate is very close to that of the best candidate?

- All proposed algorithms in the main paper provide a smoothness guarantee of C = 4. Is this by design or coincidence? If it is by design, why was C = 4 chosen, and is it possible to adjust the parameters in the algorithm to attain other trade-offs between consistency and robustness?

**Limitations:**

Yes

---

> ### Author Rebuttal · Authors · 2024-08-05
>
> We thank the reviewer for their time and effort in reviewing this paper.
>
> - “Can you provide formal comments on the definition of fairness? Is the current algorithm fair for the second-best candidate, especially when the value of the second-best candidate is very close to that of the best candidate?”
> > The formal fairness definition (see line 148) is the probability of selecting the candidate with the true maximum value. The motivation for our definition of fairness, which is based on providing guarantees only for the first-best candidate, is the following: in the offline setting where all the true values are known to the decision-maker we consider that the most deserving candidate of being selected is the first-best candidate and that the fair and optimal decision would be to select the first-best candidate with probability 1 (and the second-best candidate with probability 0, even if the true value of this candidate is very close to the true value of the best candidate). This offline setting motivates why our notion of fairness does not provide guarantees for the second-best candidate.
> That being said, we do agree that other notions of fairness would be interesting to explore. For example, as suggested by the reviewer, it would be interesting to explore a notion of fairness that provides guarantees not only for the first-best, but all candidates who are in the top-$\ell$, for some integer $\ell$, and/or those whose value is close to the value of the best candidate. We also note that our fairness definition for the $k$-secretary problem aligns with this direction.
>
>
> - “All proposed algorithms in the main paper provide a smoothness guarantee of $C = 4$. Is this by design or coincidence? If it is by design, why was $C = 4$ chosen, and is it possible to adjust the parameters in the algorithm to attain other trade-offs between consistency and robustness?”
> > $C=4$ comes from the fact that the true values of the candidate $i^*$ and $\hat{i}$ can differ by at most $2 \epsilon$ and the true value of any candidate in the set $I^{pegged}$ and $\hat{i}$ can also differ by at most $2 \epsilon$. We could change the definition of $I^{pegged}$ to include ``more exploration’’ on finding the best candidate which would lead to a larger value of $C$. However, with our current analysis, we do not see how such a change can improve the fairness constant $F$.
> Regarding other trade-offs: we can improve the fairness guarantee from $1/16 = 0.0625$ to $0.074$ and show that constant smoothness implies an upper bound of $F = 0.348$ for fairness. The proof of the latter follows from previous work: Fujii and Yoshida prove that for any constant $C$, there is no randomized algorithm with a competitive ratio better than $\max(1 − C \epsilon, 0.348)$. Since a $C$-smooth and $F$-fair algorithm has a competitive ratio of at least $\max(1 − C \epsilon, F)$, this impossibility result implies that the best achievable fairness for any $C$-smooth (where $C$ is a constant) algorithm is $0.348$.
> Exploring other trade-offs and finding the pareto-optimal curve in terms of smoothness and fairness are interesting directions. The main reason why achieving a smooth tradeoff between fairness and smoothness is challenging is the following: any bound on C for the smoothness implies a competitive ratio of $1- C \epsilon$, which implies a competitive ratio of 1 when the predictions are exactly correct. Thus, regardless of what the smoothness guarantee is, we must achieve a competitive ratio of 1 when the predictions are exactly correct, which makes it challenging to improve the fairness guarantee F, even at the expense of a worse smoothness constant C.

---

> > ### Comment · Reviewer_Le4z · 2024-08-10
> > **response to rebuttal**
> >
> > Thank you for the clarifications and additional discussions on the multi-way trade-offs in the design. My overall rating of the paper remains the same.

---

### Official Review · Reviewer_uZ3G · 2024-07-13

**Soundness:** 3
**Presentation:** 3
**Contribution:** 3
**Rating:** 5
**Confidence:** 3

**Summary:**

This paper considers the secretary problem with predictions. The decision maker is given predicted values for all the candidates in advance. The existing algorithm by Fujii and Yoshida (2023) hires the candidate with the **expected value** at least $\max ( \Omega(1), 1-O(\epsilon) )$ times the optimal candidate's value, where $\epsilon$ is the prediction error. On the other hand, the classical secretary algorithm hires the best candidate **with probability** $\Omega(1)$. Since hiring the best one with probability $p$ yields the expected value at least $p$ times the optimal value, the probability bound is stronger than the value bound.

This paper proposes an algorithm that hires the best candidate with probability $\Omega(1)$ and achieves the expected value at least $1-O(\epsilon)$ times the optimal. For the extension of hiring $k$ candidates, the proposed algorithm hires each of the top-$k$ candidates with probability depending on $k$ and achieves the expected value at least $1-O(\epsilon)$ times the optimal. The experimental results show the proposed algorithm outperforms the existing method in various benchmark instances.

**Strengths:**

- This paper is well-organized, clearly written, and easy to follow. The proofs are involved, and most of them are deferred to the appendix, but the main idea is clearly addressed in the main body.
- The problem setting is based on the existing study by Fujii and Yoshida (2023). This paper proposes a new algorithm for this setting with the property of choosing the best candidate with a constant probability (the authors call this property ``fair''). This is an interesting improvement.
- The extension to the multiple-choice secretary problem is a good technical contribution. Fujii and Yoshida also considered this setting, but they proposed only an algorithm with $1-O(\log k/k)$-type guarantee. This paper proposes a constant-factor guaranteed algorithm, also with a property of choosing top-$k$ candidates with a constant probability.
- The experimental results clearly show the empirical superiority of the proposed algorithm.

**Weaknesses:**

- In this paper, hiring the best candidate with probability $\Omega(1)$ is called ``fair.'' Although I am not familiar with the existing literature on fairness, I do not fully understand why the authors adopt this terminology. In algorithmic studies on the secretary problem, the probability maximization and value maximization are considered as two important settings. The problem setting of this paper is a mix of these two settings; the probability maximization in the case where the predictions are inaccurate, and the value maximization in the case where the predictions are accurate. This interpretation sounds more natural than the fairness notion at least to me.
- The authors claim that fairness is motivation for improving on the existing result. For the reason mentioned above, I do not think this problem setting is very natural.
- The constants in the guarantees are not optimized, probably for theoretical simplicity. Since there is no known lower bound, the existing papers on this topic (Antoniadis et al. and Fujii and Yoshida) focused on restoring or approaching the original bound $1/e$. In terms of that aspect, this paper's bound is not close to $1/e$ and difficult to highly evaluate.

Minor comments:
- The last line of page 2 contains wrong links to the reference (matroid secretary and knapsack secretary).
- To my knowledge, the first paper that derives a constant-factor competitive ratio is the knapsack secretary paper. I think it should be mentioned somewhere.

**Questions:**

If I understand correctly, in the experimental result for the Almost-Constant instance, the proposed algorithm and Dynkin's algorithm can choose the best candidate whenever the best one arrives in the latter half (time $[1/2,1]$ for the proposed algorithm and $[1/e,1]$ for Dynkin's algorithm). However, the experimental result shows both algorithms achieve the success probability approximately 0.3 or 0.4. Could you tell me how these algorithms work in this instance?

**Limitations:**

The authors are sufficiently discussing limitations of the model and algorithm in the limitations section.

---

> ### Author Rebuttal · Authors · 2024-08-05
>
> We thank the reviewer for their time and effort in reviewing this paper.
>
> - “Although I am not familiar with the existing literature on fairness, I do not fully understand why the authors adopt this terminology.”:
> > The literature on fairness contains many different definitions of fairness. We propose one definition in the context of secretaries with predictions that is motivated by the potential unfairness caused by erroneous predictions. We consider the true best candidate to be the most deserving candidate for selection and, as mentioned in lines 55-56, we consider it unfair for this true best candidate to have no chance of being selected due to some erroneous prediction. That being said, we believe there might be other definitions of fairness that would be interesting to explore in the secretaries with predictions problem.
>
>
> - "The authors claim that fairness is motivation for improving on the existing result. For the reason mentioned above, I do not think this problem setting is very natural.":
> > An alternate motivation for deriving our result is that the negative result in Fujii-Yoshida is also an upper bound on what we call “Fairness”, i.e. an upper bound on the probability of accepting the best candidate. By contrast, their algorithm only has a non-trivial lower bound on the competitive ratio in terms of valuations, and in fact (as we show in Appendix A.1) can have zero probability of accepting the best candidate.  Our result was motivated by their paper, and one of our initial goals was bridging this gap between their upper and lower bounds, which we view as a theoretically natural question.
>
>
> - “The constants in the guarantees are not optimized, probably for theoretical simplicity. The existing papers on this topic (Antoniadis et al. and Fujii and Yoshida) focused on restoring or approaching the original bound $1/e$. In terms of that aspect, this paper's bound is not close to $1/e$ and difficult to highly evaluate.”:
> > As explained in our previous response, we are looking to guarantee
> $$P[A = i^*]  \geq  F \quad (I)$$
> for some constant $F$, which is stronger than guaranteeing
> $$E[u_A] \geq F \cdot u_{i^*}\quad (II)$$
> (as done in Fujii-Yoshida).  We believe that a priori, it was unclear how to even achieve a constant $F$ for (I) while preserving 1-consistency (optimal competitive ratio when the predictions are exactly correct).  This is why we did not focus on optimizing constants, although with a bit of work, we are able to increase our guarantee on $F$ from $1/16=0.0625$ to $0.074$.  We will include this improved bound in the updated version. While we acknowledge that this ratio is still much smaller than the $0.215$ from Fujii-Yoshida, we emphasize that constants are much harder to achieve for (I) than for (II).
> We believe that achieving the optimal guarantees for this problem and characterizing the pareto-optimal curve in terms of smoothness and fairness is an interesting and challenging open direction.
>
> - “The last line of page 2 contains wrong links to the reference (matroid secretary and knapsack secretary)” and “To my knowledge, the first paper that derives a constant-factor competitive ratio is the knapsack secretary paper. I think it should be mentioned somewhere.”:
> > Thank you; we will fix the references as follows: the matroid secretary was introduced by Babaioff et al.in SODA 2007 (Matroids, secretary problems, and online mechanisms) and the knapsack secretary was introduced in [7].
>
> - “If I understand correctly, in the experimental result for the Almost-Constant instance, the proposed algorithm and Dynkin's algorithm can choose the best candidate whenever the best one arrives in the latter half (time $[1/2,1]$ for the proposed algorithm and $[1/e,1]$ for Dynkin's algorithm). However, the experimental result shows both algorithms achieve the success probability approximately $0.3$ or $ 0.4$. Could you tell me how these algorithms work in this instance?”
> > For Dynkin the probability of selecting the highest true value candidate is around $0.37 \simeq 1/e$. A necessary condition for that to happen is that this candidate arrives between times $[1/e,1]$. However that condition is not sufficient as the second highest true value candidate could also be selected if it arrives before the highest and after time $1/e$. The $0.37$ probability comes from a delicate analysis of all such events and we defer to Dynkin's algorithm analysis in [29].
> For our algorithm the probability of selecting the highest true value candidate varies from $0.33$ to $0.34$. In the following we remind that to break ties randomly but consistently we add to all predicted and true values a random noise which we make arbitrarily small so as to not interfere with the performance calculation. (see the function generate_data_sets in our code for further details).
> While it may be difficult to find a closed form solution to this probability, similarly to Dynkin’s algorithm, the highest true value candidate arriving in $[1/2,1]$ is not a sufficient condition to be selected. Indeed, if the second true value highest candidate arrives after $1/2$ and before the highest true value then it may be selected by case 4 of our algorithm as predicted and true values of all candidates except the highest true value are equal to $1$ (modulo the small noise that we add to break ties).

---

> > ### Comment · Reviewer_uZ3G · 2024-08-08
> >
> > Thank you for the feedback. The authors appropriately answered my questions. Since the lower bound $0.074$ is still far from the upper bound $0.215$ and the standard $1/e$, it is difficult for me to strongly support the acceptance of this paper, but I believe the quality of this paper is very high and above the borderline of NeurIPS.
> >
> > > In the following we remind that to break ties randomly but consistently we add to all predicted and true values a random noise which we make arbitrarily small so as to not interfere with the performance calculation.
> >
> > This resolves my concern for the experimental results. I could not find any description about adding random noise in the dataset description. If there is space to write this information, I recommend adding this.

---

### Official Review · Reviewer_3Rre · 2024-07-16

**Soundness:** 3
**Presentation:** 3
**Contribution:** 4
**Rating:** 8
**Confidence:** 4

**Summary:**

This paper examines the secretary problem with predictions and identifies a key shortcoming in prior work.  This problem is similar to the classic secretary problem in which a series of candidates with arbitrary unknown utilities $u_i$ arrive in a random order to be interviewed (revealing their utility upon arrival), and a decision maker must make an accept-or-reject decision without knowledge of the utility of future arrivals.  For the problem considered in this paper, the decision maker also has access to predictions $\hat u_i$ of the candidates' utilities.  Prior work gave algorithms guaranteed to accept a candidate with utility at least $\max(\Omega(1), 1- O(\epsilon))$ times the highest utility in expectation, where $\epsilon$ is a notion of error in the predictions.  When $\epsilon$ is very small, this can potentially be more desirable than the classical secretary algorithm which accepts the best candidate with probability $1/e$.  While this guarantee is achieved by the prior work and desirable for the decision maker, this paper shows that they do not accept the best candidate with constant probability (in fact.  This can be seen as highly unfair to the candidate with the highest utility.  The first main result of this paper is an algorithm that satisfies both types of guarantees.  In addition to this, an extension to the $k$-secretary problem and an experimental evaluation are considered.

**Strengths:**

This paper identifies an interesting and overlooked issue for the secretary problem with predictions, giving an elegant solution to it, for both the single-choice and multiple-choice secretary problems with predictions, and validates this using the experimental setup from prior work [25].  The presentation is overall clear.

**Weaknesses:**

Upper bound (impossibility results) are not explored, making it unclear if these are the tightest results for this setting.  The experiment section only addresses the single-choice secretary problem (although this is minor since the experiments already make a strong point).

**Questions:**

If we require $1-C\epsilon$-smoothness, does that imply an upper bound on the achievable $F$ for fairness, and vice-versa?

**Limitations:**

The authors have adequately discussed limitations regarding their definition of fairness.

---

> ### Author Rebuttal · Authors · 2024-08-05
>
> We thank the reviewer for their time and effort in reviewing this paper.
>
> - “Upper bound (impossibility results) are not explored, making it unclear if these are the tightest results for this setting” and “If we require $1 - C\epsilon$ smoothness, does that imply an upper bound on the achievable $F$ for fairness, and vice-versa?”:
> > Our main goal was to achieve constant-smoothness and constant-fairness and we did not focus on optimizing the constants. Thus we believe that the constants can be improved. Indeed our algorithm is 4-smooth and 0.0625-fair and with some further optimizations, we can achieve fairness with parameter 0.074.
> In terms of impossibility results, we can show that constant smoothness implies an upper bound of $F = 0.348$ for fairness. The proof follows from previous work: Fujii and Yoshida prove that for any constant $C$, there is no randomized algorithm with a competitive ratio better than $\max(1 - C\epsilon, 0.348)$. Since a $C$-smooth and $F$-fair algorithm has a competitive ratio of at least $\max(1 - C\epsilon, F)$, this impossibility result implies that the best achievable fairness for any $C$-smooth (where $C$ is a constant) algorithm is $0.348$. We will include this in the paper, along with a discussion on impossibility results and future research directions.

---

> > ### Comment · Reviewer_3Rre · 2024-08-07
> >
> > Thank you for your response and answering my question about trade-offs between smoothness and fairness.  My overall evaluation remains the same.

---

### Official Review · Reviewer_3Mim · 2024-07-29

**Soundness:** 4
**Presentation:** 2
**Contribution:** 4
**Rating:** 6
**Confidence:** 3

**Summary:**

This paper proposes new algorithm for the classical secretary problem in the algorithms with predictions framework. In particular, the paper introduces and tackles a new notion of fairness for this problem: the best candidate must have some probability of being accepted. The authors demonstrate how existing algorithms for this problem in the algorithms with predictions setting might lead to outcomes where the best candidate has zero probability of being accepted, and go on prove results on "fairness" and competitive ratio of their proposed methods Specifically, their algorithm (called Pegging), keeps a constant approximation ratio while also ensuring constant fairness. The paper also includes some experiments comparing to existing methods on synthetic examples.

**Strengths:**

The biggest strength of this work is its originality; it is one of the first works in the area of algorithms with predictions to study fairness of these algorithms. This work has the potential to spur up similar interesting questions for other existing problems/algorithms in the literature.   The key ideas are simple and the analysis is sound.

**Weaknesses:**

The main area for improvement is presentation. For instance, the key ideas behind the main algorithm are hard to understand and the reader is left trying to infer them by reading the pseudocode presented in Algorithm 1.

**Questions:**

1) Can you please define I_pegged formally? It is never defined and only mentioned at a high level in the preceding text. The reader is left confused when they reach case 1 of Algorithm 1, and ends up having to infer what this set is from case 2b (aka how are you deciding what the set of elements that would guarantee smoothness is). It is always better to help the reader.

2) Can you clarify the refine and clarify initial conditions for Algorithm 1? It would greatly help with readability if the algorithm presentation followed the standard format with input to the algorithm, initial values, and return. For instance, initialize I_pegged to empty set. Somewhat pedantic, but the while loop is ill-defined with the condition 'agent i arrives at time t_i'; for t in 1...T do ... with returns is a lot more sensible.

3) In Figure 1, why does the competitive ratio of Dynkin go down with increasing epsilon? Shouldn't it be independent of prediction error?

4) I understand there might be space limitations, but it might be illuminating to have a simple example of unfairness of an existing methods at the end of Section 2 or early in Section 3 to better shed light on the shortcomings (instead to relegating everything to the appendix). Also, minor typos at the top of Page 3 incorrectly referencing [3] instead of [4].

**Limitations:**

Yes, limitations have been addressed.

---

> ### Author Rebuttal · Authors · 2024-08-05
>
> We thank the reviewer for their time and effort in reviewing this paper.
>
> - "Can you please define I_pegged formally?" and "clarify initial conditions for Algorithm 1?":
> > $I^{pegged}$ should have been initialized to the empty set before the start of the while loop. We will fix that omission. With this initialization and subcase 2b in Algorithm 1, $I^{pegged}$ is then defined formally.
>
> - "the while loop is ill-defined with the condition 'agent i arrives at time t_i'; for t in 1...T do ... with returns is a lot more sensible.":
> > We used the model where candidates have arrival times $t_i \in [0,1]$ to simplify the analysis. As mentioned in lines 164-166, that model is equivalent to the model where candidates arrive in a uniform random order. We agree that the algorithm would be easier to read with your suggestions; thank you. We will introduce a random permutation $\sigma(t)$ mapping time steps $t \in 1 … n$ to indices of candidates and update the algorithm to have, as suggested, a for loop over $t \in 1 … n$ as well as return statements.
>
>
> - "why does the competitive ratio of Dynkin go down with increasing epsilon? Shouldn't it be independent of prediction error?":
> > In general, the competitive ratio of Dynkin should indeed be independent of the prediction error. However, in this experimental setting, the construction of the instance itself depends on the prediction error, which is why the performance of all algorithms (including Dynkin) depends on the prediction error. We will make this clearer in the future. We also emphasize that this experimental setting is replicated from the earlier work of Fujii-Yoshida as we aim to compare our algorithm to theirs.
>
>
> - "it might be illuminating to have a simple example of unfairness of an existing methods at the end of Section 2":
> > We expand on why previous algorithms lead to unfair outcomes in Appendix A but we acknowledge that a short paragraph at the end of Section 2 would improve our paper’s readability. Thanks for the suggestion; we will add such a paragraph.
>
>
> - "minor typos at the top of Page 3 incorrectly referencing [3] instead of [4].":
> > Thanks for catching the wrong reference; we will fix it!

---

> > ### Comment · Reviewer_3Mim · 2024-08-13
> >
> > Thanks for the clarifications! I will continue to recommend acceptance!

---

### Decision · Program_Chairs · 2024-09-25

**Decision:**

Accept (poster)

**Comment:**

This paper studies algorithms for the well-known secretary problem in an optimizing with predictions framework, focusing mostly on providing theoretical analysis and guarantees and with some empirical demonstration. Reviewers found the paper to be novel and interesting, appreciated the consideration of fairness concerns and its particular manifestation in the proposed problem setup (although the authors are encouraged to make precise their notion of fairness as it relates to how "fairness" is generally thought of currently in the learning community), and found the results to be relevant and useful. Overall the paper is clearly written, well executed, and will likely inspire future interest in this problem (e.g., improved constants, upper bound) and in related problems.